# Deglacial mobilization of pre-aged terrestrial carbon from degrading permafrost

Maria Winterfeld[1,2], Gesine Mollenhauer [1,2,3], Wolf Dummann[2,7], Peter Köhler [1], Lester Lembke-Jene [1] Vera D. Meyer[1], Jens Hefter[1], Cameron McIntyre [4,8], Lukas Wacker[5], Ulla Kokfelt[6] & Ralf Tiedemann[1,2]

The mobilization of glacial permafrost carbon during the last glacial–interglacial transition has been suggested by indirect evidence to be an additional and significant source of greenhouse gases to the atmosphere, especially at times of rapid sea-level rise. Here we present the first direct evidence for the release of ancient carbon from degrading permafrost in East Asia during the last 17 kyrs, using biomarkers and radiocarbon dating of terrigenous material found in two sediment cores from the Okhotsk Sea. Upscaling our results to the whole Arctic shelf area, we show by carbon cycle simulations that deglacial permafrost-carbon release through sea-level rise likely contributed significantly to the changes in atmospheric $CO_2$ around 14.6 and 11.5 kyrs BP.

[1] Alfred-Wegener-Institut, Helmholtz-Zentrum für Polar-und Meeresforschung (AWI), 27570 Bremerhaven, Germany. [2] Department of Geosciences, University of Bremen, 28359 Bremen, Germany. [3] MARUM—Center for Marine Environmental Sciences, University of Bremen, 28359 Bremen, Germany. [4] Department of Earth Sciences, Geological Institute, ETH Zürich, 8092 Zürich, Switzerland. [5] Department of Physics, Laboratory for Ion Beam Physics, ETH Zürich, 8093 Zürich, Switzerland. [6] Geological Survey of Denmark and Greenland, 1350 Copenhagen, Denmark. [7] Present address: Institute of Geology and Mineralogy, University of Cologne, 50674 Cologne, Germany. [8] Present address: Scottish Universities Environmental Research Centre, East Kilbride G75 0QF, UK. Correspondence and requests for materials should be addressed to G.M. (email: gesine.mollenhauer@awi.de)

The carbon reservoir in northern high-latitude permafrost regions is substantially larger than the carbon content of the atmosphere today, and has been estimated to be approximately 1.5 times larger than its modern size during the Last Glacial Maximum (LGM)[1,2]. Consequently, permafrost degradation and rapid respiration of previously frozen organic matter during the last deglaciation potentially had profound effects on the global carbon cycle via positive feedbacks. Growing evidence suggests that large quantities of pre-aged carbon released from degrading permafrost contributed to the deglacial rise in atmospheric carbon dioxide and methane concentrations, thereby amplifying warming[1,3,4]. The deglacial processes might serve as modern analogues, as positive climate feedbacks from coastal erosion and hinterland thawing of carbon-rich permafrost soil deposits are also expected under future warming[5,6]. The rates and pathways of carbon release from permafrost are highly uncertain, but crucial to understand how strongly, and over which time-scales, these feedbacks may affect climate[7].

Net losses of carbon from degrading permafrost, which has been photosynthetically fixed millennia before mobilization and thus is depleted in radiocarbon, will result in export of pre-aged carbon. Permafrost degradation processes include thawing and deepening of the active layer, extension of wetlands, development of thermokarst resulting in subsidence, lake development and land-sliding, and coastal erosion. Until now, assessments of the susceptibility of permafrost to degradation and assessments of future and past feedbacks rely mainly on simulation scenarios, including assumptions based only on indirect estimates of the age of permafrost-derived old carbon, but physical data are so far lacking[3,6]. Thus, the age, timing and quantity of terrigenous carbon remobilized during the last deglaciation is largely unknown.

In contrast to dissolved organic matter, particulate organic matter is the only fraction of exported material that can be preserved in sediments and records past climate and environmental conditions. Accumulation rates and radiocarbon ages of terrigenous carbon in sediments adjacent to areas of permafrost degradation thus allow a data-based evaluation of the permafrost carbon remobilization and associated carbon-climate feedback, including its contribution to deglacial $CO_2$ rise.

To provide more insights into past East Asian permafrost dynamics, we here present a reconstruction of vegetation development in, and terrestrial carbon accumulation off of, the Amur River catchment over the past 17 kyrs. We analysed terrestrial organic compounds and their radiocarbon ages deposited in marine sediments off the mouth of the Amur River in the Okhotsk Sea, a marginal sea of the North Pacific Ocean. The Amur River basin is the largest catchment in East Asia. The region was completely covered by permafrost during the LGM[8] and is, as a result of deglacial permafrost mobilization, almost entirely permafrost-free today (Fig. 1)[9]. The East Sakhalin margin near the mouth of the Amur River is a location of high sediment accumulation, where river-discharged material, together with terrigenous particles entrained in ocean currents and derived from the shelf areas of the northwestern Okhotsk Sea, accumulates[10] (see Methods). We studied material from two sediment cores, LV28-4-4 (51°08.475′ N, 145°18.582′ E; 674 m water depth) and SO178-13-6 (52°43.881′N, 144°42.647′E; 713 m water depth), retrieved from this zone of high glacial to Holocene sedimentation. We analysed concentrations of two terrigenous biomarker groups (high molecular weight (HMW) n-alkanes, branched glycerol dialkyl glycerol tetraethers (brGDGTs)) and quantified the age of the deposited terrigenous material by compound-specific radiocarbon dating of HMW n-alkanoic acids. In our analysis, we compare our accumulation rates of terrigenous biomarkers with rates of global sea level rise, wetland development

and changes in Amur river discharge, and the pre-depositional age of terrigenous biomarkers with deglacial records of the carbon cycle from the literature[11–13]. Thereby, we investigated mobilization of pre-aged terrestrial carbon by coastal erosion and thermal permafrost degradation in the hinterland and subsequent fluvial export. By upscaling and applying our new findings as boundary conditions to a carbon cycle model, we simulate how this mobilization of permafrost carbon might have influenced the levels of atmospheric $CO_2$ and $\Delta^{14}C$. We find that about 50% of the abrupt rises in atmospheric $CO_2$ found in ice cores at 14.6 and 11.5 kyrs BP can be explained by this mobilization of pre-aged terrestrial carbon via coastal erosion in the Arctic Ocean following sea-level rise. However, the long-term contributions of this process to the $CO_2$ rise and $\Delta^{14}C$ decline across Termination I are small.

## Results

**Terrigenous material accumulation in Okhotsk Sea sediments.** Highest accumulation rates of HMW long-chain ($C_{27}$–$C_{33}$) n-alkanes derived from terrestrial higher plants, and soil microbial brGDGTs are observed during the deglaciation between 17 and 10 kyrs BP, decreasing towards the Holocene and reaching low and rather constant values after 8 kyrs BP (Fig. 2c, d). Two distinct maxima are observed in both biomarker records, namely between 14.5 and 13 kyrs BP during the Bølling-Allerød (B/A), and between 11.5 and 10 kyrs BP during the Pre-Boreal periods. Furthermore, a less-pronounced local maximum exists between 17 and 15.5 kyrs BP during Heinrich Stadial 1 (HS1).

These records of terrigenous material accumulation reach their highest values during times of rapid sea-level rise (Fig. 2b)[14], connected with the global melt water pulses (MWP) centred around 11 and 14 kyrs BP raising sea-level by a total of approximately 80 m. We interpret this temporal coincidence as an indication that coastal erosion was the main cause for this permafrost carbon mobilization. The abrupt increase in biomarker accumulation at these times implies that erosion of coastal permafrost happened relatively fast, which is plausible, as today coastal erosion of permafrost deposits occurs at extremely rapid rates[15] and results in failure of coastal bluffs, supplying large amounts of particulate organic matter directly to the ocean[16]. In a next step, this material is then distributed, further degraded, and re-buried in marine sediments[5,17].

The deglacial warming also led to permafrost degradation within the Amur catchment, likely resulting in the formation of thermokarst lakes and wetland expansion, followed by drying and further thawing[18]. Analogous to today, permafrost-derived carbon likely has been released via emissions of $CO_2$ and $CH_4$ into the atmosphere[6,7,19], and via dissolved and particulate organic matter through rivers to the ocean[20]. Being the dominant aquatic carbon export pathway, dissolved organic matter derived from thawing permafrost today is rapidly oxidized in Arctic rivers[21] and significantly contributes to greenhouse gas emissions. The hydrological evolution and resulting vegetation development is thus tightly linked to the process of ongoing permafrost retreat. The Amur river discharge as reconstructed from the accumulation rate of freshwater algae (Fig. 2f, Methods), was considerably higher during the deglaciation than it is today. It reached its maximum around 10 kyrs BP (Fig. 2f) likely exporting vast amounts of dissolved and particulate organic matter during this time interval, which is indicated by relatively higher biomarker accumulation rates between 10 and 8 kyrs BP compared to the modern situation (Fig. 2c, d). However, because of the different timing of the Amur discharge maximum and the peaks in biomarker accumulation rates centred at ~14 and 11 kyrs BP, especially during the Pre-Boreal, we conclude that river-derived

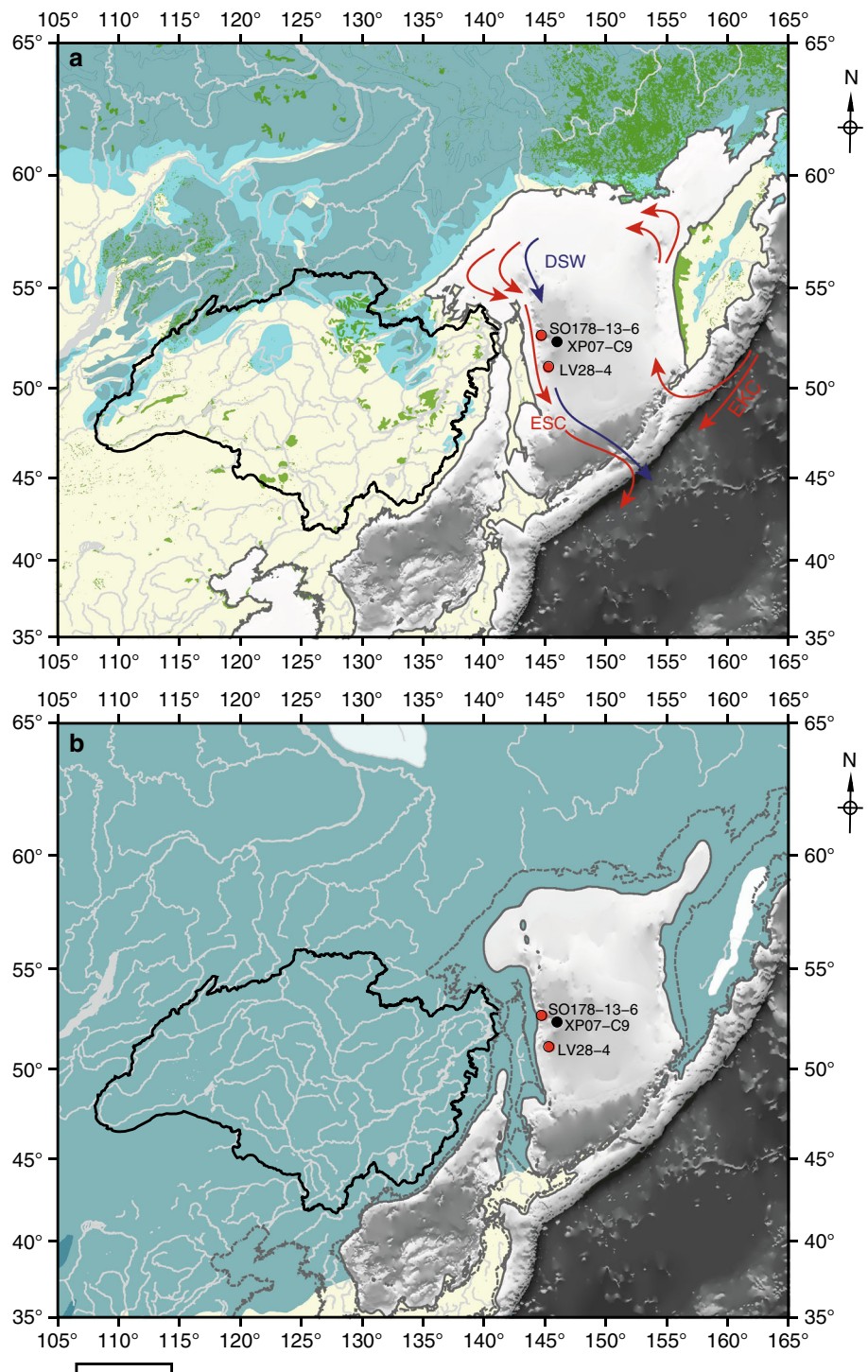

**Fig. 1** Permafrost distribution in East Asia. Okhotsk Sea study area with locations of the investigated cores (red circles) and a core referenced in this study[26]. The Amur River basin is outlined in black. **a** Modern permafrost extent[66] is indicated in blue (dark: continuous permafrost, light: discontinuous and sporadic permafrost) and wetlands[67] in green. Red arrows represent the surface water circulation with the East Sakhalin Current (ESC) and the East Kamchatka Current (EKG), and blue arrows represent the Dense Shelf Water (DSW) pathways. **b** Permafrost extent[8] and exposed shelf areas (132 m isobaths) during the last glacial (~21 kyrs BP). The modern coastline is indicated by the dashed line. Maps are created using GMT (Generic Mapping Tools, http://gmt.soest.hawaii.edu/) software

organic matter has only partially contributed to these peaks. Since the Amur discharge is largely controlled by precipitation intensity in East Asia, i.e., the East Asian summer monsoon (Fig. 2g)[22], we find a general synchronicity of river discharge and monsoonal precipitation, which follows local summer insolation, as recorded in speleothems[23] (Fig. 2g) in the early Holocene.

**Age of terrigenous organic matter**. Compound-specific radio-carbon analyses (CSRA; Methods) of HMW $n$-alkanoic acids ($C_{26:0}$ and $C_{28:0}$) from selected intervals (six from SO178-13-6 and four from LV28-4-4) reveal that during the deglaciation terrigenous biomarkers were strongly pre-aged by 5 to 10 kyrs at the time of deposition and became progressively younger after the

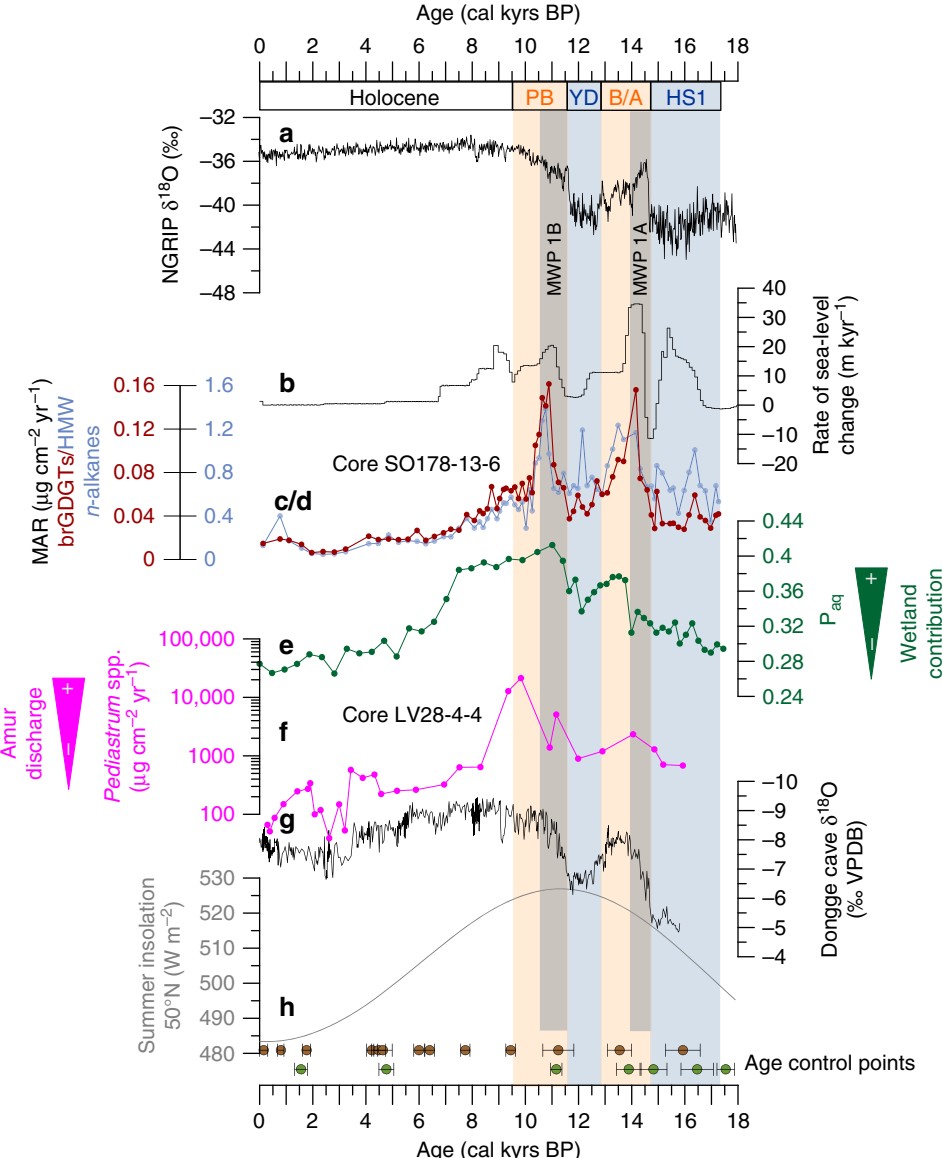

**Fig. 2** Proxies for terrigenous organic matter mobilization compared with records of deglacial environmental changes. **a** Greenland NGRIP δ[18]O[68]; **b** rate of global sea-level change[14]; biomarker records obtained from sediment core SO178-13-6, mass accumulation rate (MAR) of **c** branched glycerol dialkyl glycerol tetraethers (brGDGTs) and **d** high molecular weight (HMW) n-alkanes ($C_{27}$, $C_{29}$, $C_{31}$ and $C_{33}$); **e** $P_{aq}$ ratio[25] from nearby core XP07-C9[26]; **f** accumulation rate (AR) of chlorophycean freshwater algae *Pediastrum* spp. from core LV28-4-4, note the logarithmic scale; **g** speleothem δ[18]O record from Dongge cave as indicator for East Asian summer monsoon intensity, which controls precipitation in the Amur Basin[23]; **h** summer insolation at 50°N. Circles at the bottom show age control points (AMS [14]C dates) with 2σ uncertainties for cores LV28-4-4 (brown) and SO178-13-6 (green)[46]. Orange boxes highlight the warm phases Bølling-Allerød (B/A) and Pre-Boreal (PB), the Younger Dryas cold spell (YD) and Heinrich Stadial 1 (HS1) are marked in blue; grey boxes mark the periods of melt water pulse 1 A (MWP-1A) and MWP-1B

Pre-Boreal towards the late Holocene (Table 1, Fig. 3c; late Holocene dates are not shown in Figure). These pre-depositional ages indicate that terrestrial deposits, which were eroded during deglacial sea-level rise and exported from degrading permafrost soils within the Amur catchment, contained large amounts of several millennia-old carbon, a situation which is comparable to modern-day organic matter-rich permafrost deposits, particularly the ice-rich Yedoma found as glacial relict primarily in eastern Siberia, Alaska, and parts of Canada (ref. [24] and references therein). Such types of deposit likely covered large areas of the glacial Asian continent and adjacent exposed shelves[18], and its degradation and erosion led to massive terrestrial organic carbon remobilization and partial burial in marine sediments. Today,

erosion of Yedoma along the coasts results in complete failure of the coastal bluffs[16], and there is no evidence for Yedoma pre-served subsea. The material eroded from Yedoma settles rapidly, resulting in high burial rates of strongly pre-aged terrestrial carbon in near-coast and shelf sediments[5,17]. Terrigenous HMW n-alkanoic acids deposited during peak discharge of the Amur at around 10 kyrs BP and also later at around 8 kyrs BP are likewise pre-aged, ~8–10 kyrs old at the time of deposition (Fig. 3c), indicating that also the terrestrial carbon transported from the thawing hinterland permafrost by the Amur river to the Okhotsk Sea contained several millennia old carbon and thus contributed to the accumulation of pre-aged terrestrial carbon in the sediment.

**Table 1 Compound-specific radiocarbon data ± propagated errors (σ) of long-chain n-alkanoic acids for cores SO178-13-6 and LV28-4-4**

| Sample depth (cm below surface) | Deposition age (mid-point) [cal kyrs BP] | n-alkanoic acid | Corrected F$^{14}$C ± σF$^{14}$C[a] | Δ$^{14}$C ± σ$^{14}$C [‰] | Age at deposition ± 1σ [yr] |
|---|---|---|---|---|---|
| Core SO178-13-6 | | | | | |
| 55–65 | 0.76 | n-C26:0 | 0.6465 ± 0.0064 | −358.5 ± 6.1 | 3065 ± 123 |
| 695–705 | 5.92 | n-C26:0 | 0.3078 ± 0.0041 | −694.6 ± 4.0 | 4790 ± 120 |
| 1435–1445 | 10.02 | n-C26:0[b] | 0.1567 ± 0.0036 | −844.5 ± 3.4 | 8060 ± 210 |
| 1435–1445 | 10.02 | n-C28:0[b] | 0.1514 ± 0.0035 | −849.7 ± 3.3 | 8410 ± 205 |
| 1805–1815 | 11.97 | n-C26:0 | 0.1762 ± 0.0046 | −825.2 ± 4.5 | 4980 ± 325 |
| 2033–2041 | 14.16 | n-C26:0[b] | 0.1152 ± 0.0031 | −885.7 ± 2.9 | 6780 ± 290 |
| 2335–2342 | 17.26 | n-C26:0 | 0.0683 ± 0.0030 | −932.2 ± 2.9 | 8600 ± 260 |
| Core LV28-4-4 | | | | | |
| 54–56 | 0.71 | n-C26:0 | 0.5943 ± 0.0084 | −411.1 ± 8.1 | 4115 ± 50 |
| 54–56 | 0.71 | n-C28:0 | 0.5692 ± 0.0102 | −435.2 ± 9.8 | 4580 ± 190 |
| 751–753 | 8.29 | n-C28:0 | 0.1586 ± 0.0041 | −842.6 ± 4.0 | 9690 ± 240 |
| 860–862 | 11.78 | n-C26:0 | 0.1045 ± 0.0036 | −896.3 ± 3.5 | 10,160 ± 345 |
| 860–862 | 11.78 | n-C28:0 | 0.1100 ± 0.0046 | −890.8 ± 4.4 | 9690 ± 445 |
| 926–928 | 16.0 | n-C26:0 | 0.0973 ± 0.0039 | −736.0 ± 4.2 | 6530 ± 340 |

[a]Corrected for blank contribution and methylation; ±propagated error (see Methods)
[b]66% split of sample

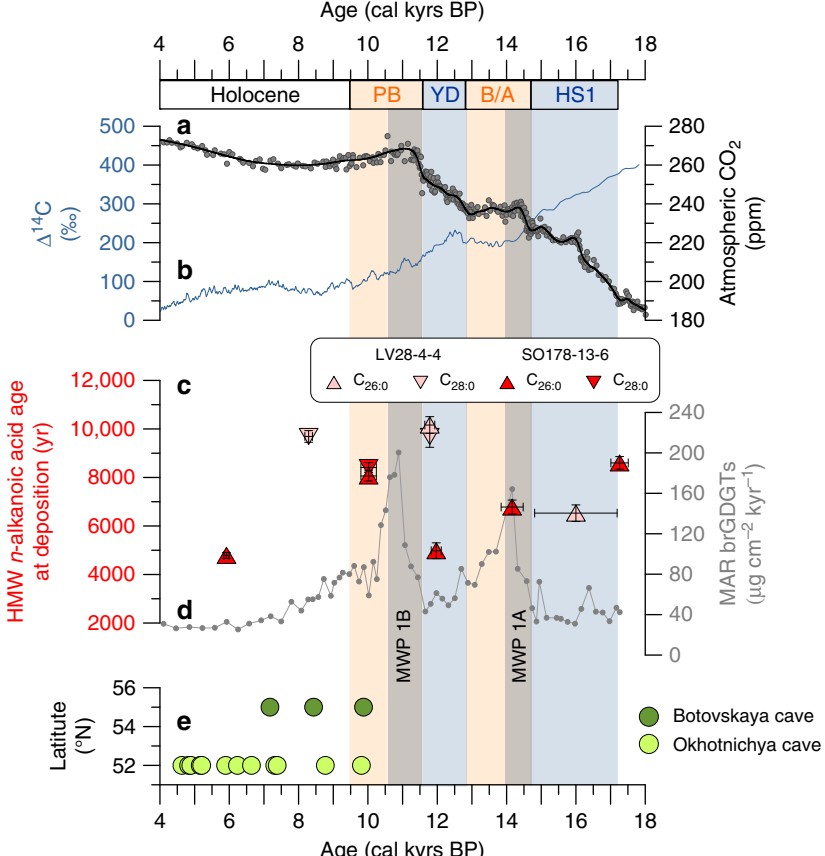

**Fig. 3** Proxies used to reconstruct permafrost dynamics and carbon mobilization. **a** Atmospheric $CO_2$ mixing ratio from ice cores[12, 13] (data as in ref. [69]); **b** atmospheric Δ$^{14}$C (‰) as reconstructed in IntCAL13[11]; **c** age at deposition of higher plant derived $C_{26:0}$ and $C_{28:0}$ n-alkanoic acids from SO178-13-6 (red triangles) and LV28-4-4 (pink triangles) derived from compound-specific radiocarbon dates of the respective biomarkers with horizontal error bars representing 2σ age uncertainties of the closest age control point (see Fig. 2) and vertical error bars representing the propagated age uncertainties after blank correction (see Methods), please note that the age scale goes from 4 to 18 kyrs BP here and that additional ages of n-alkanoic acids deposited before 4 kyrs BP can be found in Table 1; **d** mass accumulation rates (MAR) of branched glycerol dialkyl glycerol tetraethers (brGDGTs) of core SO178-13-6 representative of MAR of all terrigenous biomarkers; **e** periods of vadose speleothem growth linked to permafrost thaw and/or absence at the Botovskaya cave and the Okhotnichya cave[9]. Orange boxes highlight the warm phases Bølling-Allerød (B/A) and Pre-Boreal (PB), the Younger Dryas cold spell (YD) and Heinrich Stadial 1 (HS1) are marked in blue; grey boxes mark the periods of melt water pulse 1a (MWP-1A) and the putative MWP-1B

**Timing of terrigenous material accumulation peaks.** Our records of accumulation rates of terrigenous biomarkers and their age at the time of deposition (Fig. 3c) illustrate the relative timing and interplay of two deglacial processes—coastal erosion and hinterland thermal degradation. At approximately 17.25 kyrs BP, when the rate of sea-level rise was low, HMW $n$-alkanoic acids were about 8.6 kyrs old at the time of deposition. Compound-specific radiocarbon ages of HMW $n$-alkanoic acids in modern core-top sediments of the East Siberian Shelf reach values of up to 18 kyrs at locations of active coastal erosion of glacial deposits[17]. Thus, the reconstructed pre-depositional age of 8.6 kyrs is comparable to the modern situation when accounting for the additional ageing of >10 kyrs since the organic matter was buried at our core locations during the last deglaciation. During the period starting slightly before and encompassing MWP 1A, which was also characterized by slightly increasing river discharge, HMW $n$-alkanoic acids were about 6.5 kyrs in age at the time of deposition, and terrigenous biomarkers accumulated at high rates. These deposits might have been partly derived from coastal erosion and from inland material, the latter of which is found to be between 3 and 5 kyrs old in the Arctic Lena river delta today[17]. Synchronous to the phase of lowest deglacial sea-level rise, HMW $n$-alkanoic acids of at least 5 kyrs in age were deposited during the Younger Dryas cold period. During the Pre-Boreal, a second peak in terrigenous biomarker accumulation occurred, and HMW $n$-alkanoic acids deposited before this peak were ~10 kyrs old at deposition. Likely, this event was again primarily related to sea-level rise induced erosion, as the onset in our MAR is delayed from the abrupt rise in global sea level by only a few centuries (Fig. 2a–d), which might be due to regional differences in sea-level rise within the Okhotsk Sea as well as age model uncertainties of the investigated sediment core. The maximum peak in Amur river discharge (Fig. 2f) occurred later at around 10 kyrs BP, when slightly younger but still strongly pre-aged HMW $n$-alkanoic acids (~8 kyrs in age) were deposited, presumably containing a substantial fraction of material from degrading inland permafrost exported through the Amur river. This degradation of permafrost and associated wetland development during the B/A, and particularly the Pre-Boreal warm phases continuing until ~8 kyrs BP, is supported by the $P_{aq}$-record[25], representing leaf-wax lipids ascribed to aquatic plants versus lipids derived from higher vascular plants, from a nearby core (Figs. 1 and 2e)[26]. Furthermore, the generally wetter conditions in the Amur catchment during the Pre-Boreal are in agreement with prior studies of pollen and biomarker records in peatlands[27,28]. These wetlands located at the southern boundary of the boreal region are potential source areas of $CH_4$, which could have contributed to the rapid deglacial increases in atmospheric methane[29]. By the onset of the Holocene the continuous permafrost boundary almost reached its modern position leaving most of the Amur catchment permafrost-free, which is indicated by speleothem growth initiation in two caves near Lake Baikal on the northeastern boundary of the Amur catchment (Fig. 3e)[9].

## Discussion

Our records indicate a contribution of degrading permafrost to atmospheric $CO_2$ and $\Delta^{14}C$ levels, the magnitude of which we attempt to estimate based on an extrapolation of our results using a carbon cycle model. On a global scale, the shelf areas of the Okhotsk Sea are relatively small, implying that their flooding alone could not explain a significant contribution to the rise in atmospheric $CO_2$ concentration. The Bering, Chukchi, and East Siberian Shelves (together with the shelf areas of the Okhotsk Sea summarized as Arctic shelves from here on) were likely flooded at around the same time, resulting in substantial emission of $^{14}C$-

depleted $CO_2$. The contribution from degrading hinterland permafrost as a result of thermokarst lake and wetland development as well as soil destabilization and erosion are spatially heterogeneous processes and therefore difficult to extrapolate to the wider permafrost region. In our extrapolation, we therefore focus in the following on the potential contribution from sea-level-induced coastal erosion. Deglacial sea-level rise eventually flooded 1.9 million $km^2$ of the Arctic shelf area[24,30]. About 80% of the exposed shelf is assumed to have been covered by Yedoma during the glacial sea-level lowstand[18]. A recent study estimated that 259 PgC were lost from the total Yedoma domain during the transition between the last glacial maximum and the Holocene[23] providing an estimate of glacial permafrost carbon that might have been deposited on the Arctic shelves and was remobilized during their flooding.

Between 18 and 11 kyrs global sea level rose from 130 to 50 m below present[14]. Considering the regional bathymetry, we calculated consistently from two different approaches (similarly as in either ref. [3] or ref. [30]), that about 50% of the Arctic shelf area was flooded during this time period by this 80 m rise in global sea level. We therefore take a similar fraction of the proposed permafrost carbon deposited on the Arctic shelves (i.e.,129.5 PgC) into account that could have been released during this time interval. For our simulation of the potential impact of permafrost carbon release, we assume, based on a recent estimate of remineralization rate of eroded Yedoma at an Arctic coast[5], that 66% of this organic matter (i.e., ~85 PgC) was oxidized to $CO_2$ and released to the atmosphere.

Since our biomarker-MAR records show three distinct peaks (Fig. 2c, d) which are related to sea level rise, we focus our simulations on the assumption that the proposed 85 PgC have been released by these three events. Furthermore, we rely on an earlier simulation study[3] discussing the 14.6 kyr-event, that clearly showed (based on U/Th-dated atmospheric $^{14}C$ anomalies from Tahiti corals)[31] that a sea level-related carbon release started no later than 14.6 kyrs BP. This information pins the start of our carbon release from flooded shelf permafrost to the onset of the sea level rise (Fig. 4a), while the peak in our biomarker MARs occurred some centuries later, potentially delayed by some transformation processes[32]. However, the apparent delay might solely be related to age model uncertainties.

Using the global carbon cycle model BICYCLE[3], we simulated how the flooding of the Arctic shelves during the last deglaciation may have contributed to changes in atmospheric carbon records. In our model simulation, $CO_2$ release from permafrost, derived from the assumption that 66% of mobilized permafrost carbon was respired, was restricted to a time window of 200 years similar to a previous study[3]. Both the release length and the pinning of its onset to sea level changes was assumed to be identical for the two other events at 16.5 and 11.5 kyrs, while the annual release rates (0.17 or 0.09 PgC per year) were derived from the total carbon amount that was assumed to be remineralized approximately scaled to the amplitudes of our biomarker MAR records. As a result, our model simulated a permafrost carbon release of 34 PgC at 11.5 and 14.6 kyrs BP and of 17 PgC at 16.5 kyrs BP (Fig. 4a). In an alternative scenario, the gradual release of the 85 PgC was simulated with a constant release rate across the last deglaciation.

Our results show that the sea-level rise-induced rapid mobilization of old permafrost-derived carbon likely coincided with the abrupt rises in the atmospheric $CO_2$ record[12] at 14.6 and 11.5 kyrs BP, but not at 16.5 kyrs BP (Fig. 5a, c, d). At 14.6 kyrs BP we simulate a $CO_2$ peak of 6 ppm (Fig. 5a) together with a drop in $\Delta^{14}C$ of 6 or 8 ‰ (Fig. 5b) depending on whether a pre-depositional carbon age of either 5 or 10 kyrs was assumed. For this event, permafrost thawing has already been suggested as a possible cause[3], but assuming greater carbon release of 125 PgC

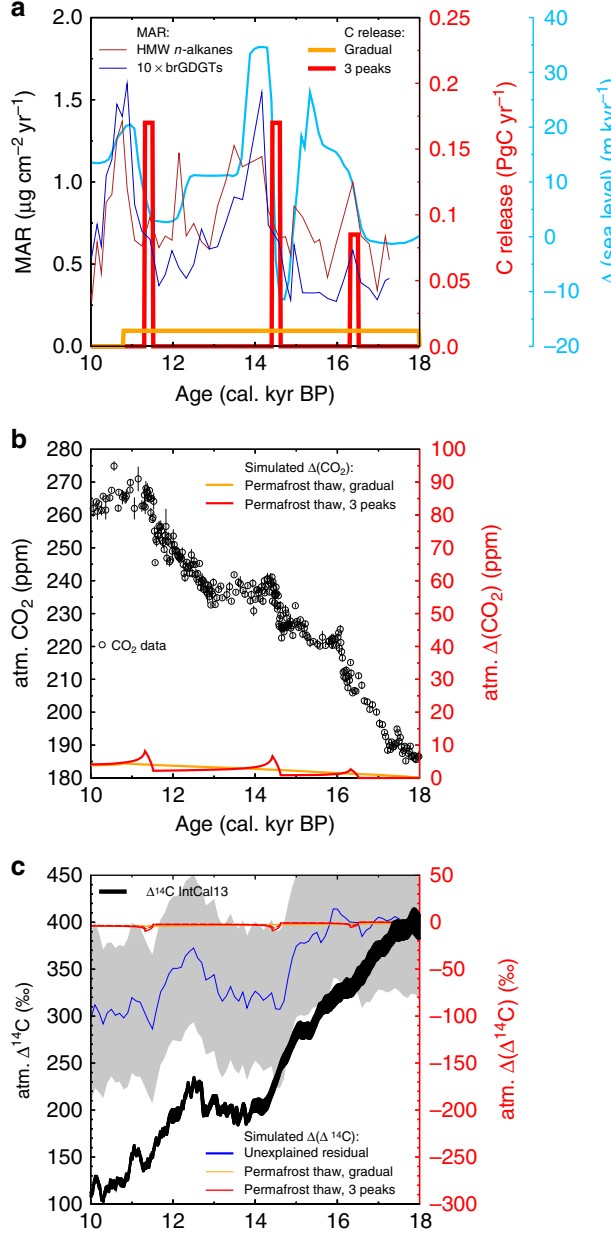

**Fig. 4** Simulated impacts of sea level triggered coastal erosion and related permafrost thawing on atmospheric carbon reservoirs using the global carbon cycle model BICYCLE. **a** Assumed carbon release from permafrost thaw of 85 PgC from 18 to 10.8 kyrs BP, either gradual (orange) or in 3 short periods of 200 yr duration connected with rapid sea level rise (red). For comparison our new MAR data and a reconstruction of sea level change[14] are also shown. **b** Simulated anomalies in atmospheric $CO_2$ levels for the two carbon release scenarios and reconstructed $CO_2$ data (mean ± 1σ) from ice cores[12, 13] for comparison (data as in ref. [69]). **c** Simulated anomalies in atmospheric $\Delta^{14}C$. The unexplained residual (mean (blue) ± 1σ uncertainty (grey)) shows $\Delta(\Delta^{14}C)$ that is not explained by changes in $^{14}C$ production rate. Anomalies in $\Delta^{14}C$ caused by the two carbon release scenarios with difference in the pre-depositional ages of the released carbon (5 kyrs: broken lines; 10 kyrs: solid lines). IntCal13[11] atmospheric $\Delta^{14}C$ for comparison

leading to a true atmospheric $CO_2$ peak of more than 20 ppm, that might have been recorded as a $CO_2$ rise of 12 ppm in about two centuries in the EPICA Dome C ice core. However, this earlier interpretation[3] was based on a $CO_2$ record with lower resolution, while newer $CO_2$ data from the WAIS Divide ice core[12] provide more constraints on the amplitude and allow, due to a refined understanding of gas enclosure processes in the WAIS Divide ice core[33], only little overshoot of the true atmospheric signal when compared to the ice core-based $CO_2$ rise. The full $CO_2$ amplitude of the 14.6 kyr-event in WAIS Divide ice core was calculated to be $12 \pm 1$ ppm when averaging data points before and after the rise[12], but might actually be around 15 ppm when calculating the peak-to-peak-difference (Fig. 5a). Such a $CO_2$ peak would be explained in our model by the release of the entire estimate of 85 PgC within only two centuries. This amount of carbon is significantly lower than in the initial proposal[3], but if solely based on the radiocarbon-depleted carbon from permafrost thawing, might still explain the $\Delta^{14}C$ anomaly in the Tahiti corals (Fig. 5b). Terrestrial biomarker records in a sediment core retrieved in the Black Sea[34] also point towards the degradation of permafrost in Eastern Europe at the onset of the Northern Hemisphere warming into the Bølling-Allerød, potentially contributing to the rapid $CO_2$ rise at 14.6 kyrs BP.

The simulated amplitudes for both the 11.5 and 14.6 kyr-event are similar, since they are based on identical carbon release rates, and would explain a substantial part of the $CO_2$ rise found in the WAIS Divide ice core[12], which has been quantified to $13 \pm 1$ ppm at 11.5 kyrs BP (Fig. 5c). In line with our baseline assumptions, evidence for permafrost carbon mobilization during this time period has also been found on the Laptev Sea shelf[20].

At 16.5 kyrs BP, we simulated a rise in $CO_2$ of only 3 ppm and a decline in $\Delta^{14}C$ of 4–6‰ (Fig. 4). Here, the simulated $CO_2$ rise occurs 300 years earlier than the abrupt rise seen in the WAIS Divide ice core and is only a quarter of the amplitude in the ice core data (Fig. 5d). Based on the published chronologies and our assumption of a synchronicity of the onset of terrestrial carbon releases and abrupt sea level rises supported by the U/Th dated $^{14}C$ available for the 14.6 ka event, it seems unlikely that at 16.5 kyrs BP the rapid $CO_2$ rise is related to sea-level induced permafrost erosion. However, future improvements of age model uncertainties might support different conclusions.

Our model-based extrapolations of carbon release from permafrost thawing and degradation through coastal erosion on the global carbon cycle are a rough first estimate, which needs further support from independent data. Admittedly, some uncertainties exist in our assessment, e.g. the simulated atmospheric carbon anomalies are model-dependent, and our model has a rather small airborne fraction when compared to others (Supplementary Fig. 1a), implying that carbon released into the atmosphere is quickly taken up by the ocean. Models with higher airborne fractions would therefore simulate larger $CO_2$ amplitudes based on the amounts of released carbon we estimated in our study. The extrapolated amount of carbon on the Arctic shelf also contains uncertainties on the order of 50% in the estimate of the Yedoma organic carbon content. Furthermore, some of the permafrost-derived carbon is found in our sediment cores in the Okhotsk Sea, underlining the uncertainties associated with the estimated remineralization rate of 66%, which is potentially too high. The current range of estimates for carbon loss upon thaw varies from 2 to ~80% of the initial carbon stock[35,36].

Constraints on the underlying processes responsible for the abrupt rises in atmospheric $CO_2$ have been provided by $\delta^{13}C$ analyses of $CO_2$[2,37]. It has been concluded that terrestrial carbon release alone cannot fully account for the atmospheric $CO_2$ increase at 14.6 and 11.5 kyrs BP, which would have led to a drop in atmospheric $\delta^{13}C$-$CO_2$. In agreement with the results from our

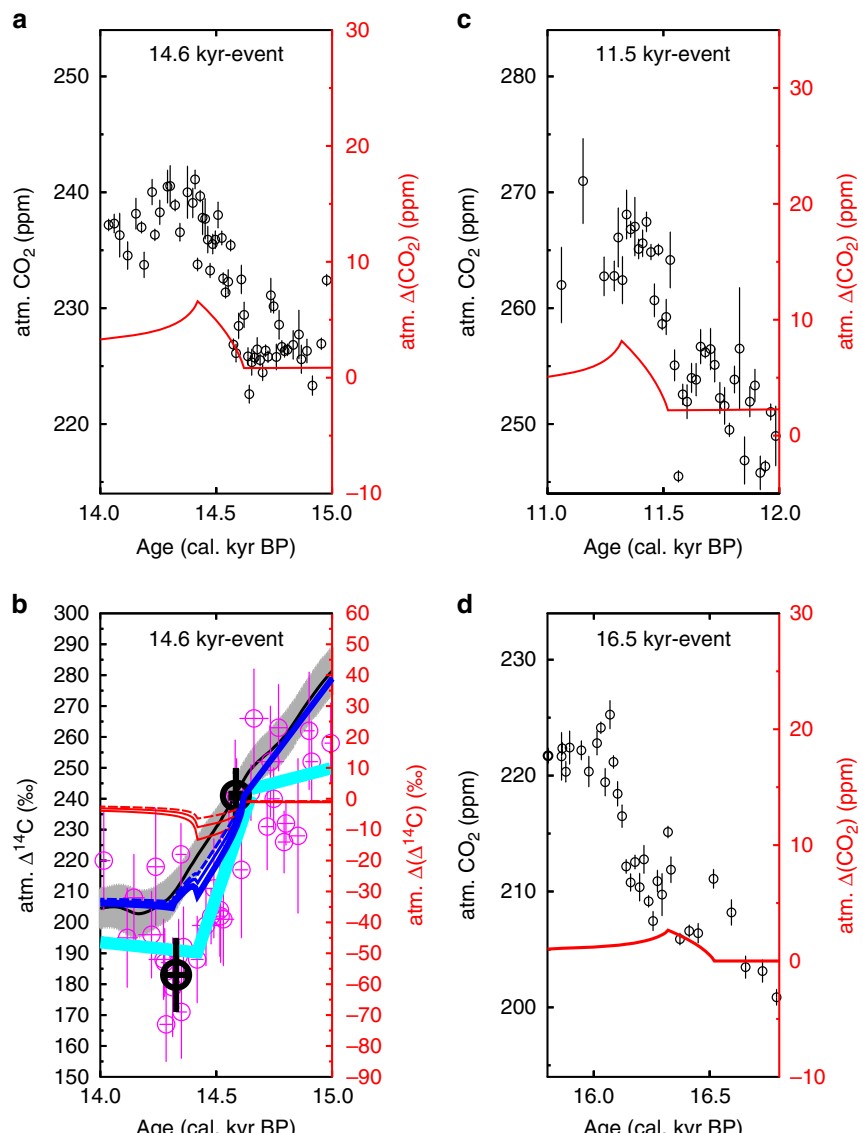

**Fig. 5** Zoom-in on proposed 3 events of coastal erosion-based carbon cycle changes. $CO_2$ changes for **a** 14.6 kyr-event, **c** 11.5 kyr-event, **d** 16.5 kyr-event. Circles are $CO_2$ data from ice cores (refs. [12,13], data as in ref. [69]). **b** Radiocarbon impacts during 14.6 kyr-event. IntCal13 (black with grey uncertainty band). High-resolution U/Th-dated $\Delta^{14}C$ from Tahiti corals (magenta)[31]. Linear change in the Tahiti-based data $^{14}C$ calculated with the Breakfit software (cyan bold line)[70] and mean of Tahiti $\Delta^{14}C$ data before and after break (bold black circles). Simulated $\Delta(\Delta^{14}C)$ based on 5 (red broken) and 10 (red thin) kyrs pre-depositional aged carbon and for a scenario with prescribed $\Delta(\Delta^{14}C)$ of $-1250‰$ (red thick) potentially possible from radiocarbon free $CO_2$ (pre-depositional age > 30 kyrs). Alternatively, a background trend in $\Delta(\Delta^{14}C)$ of $-0.1‰$ per year was added to the scenarios (blue lines). All uncertainties are ± 1σ

modelling exercise, this indicates that these $CO_2$ peaks are probably difficult to explain with permafrost degradation alone, but rather suggest a combination of terrestrial and marine processes occurring simultaneously. Furthermore, both events took place at the same time as an abrupt warming in the Northern Hemisphere indicating a change in the bipolar seesaw[38]. Consequently, some ocean circulation changes are also expected to have occurred, potentially obscuring terrestrial-based carbon release[39].

Since permafrost thawing and degradation including thermokarst lake and wetland development likely result in outgassing of both $CO_2$ and $CH_4$, these processes would potentially have contributed to the concurrent rise in $CH_4$ at 11.5 kyrs BP[12]. However, radiocarbon analyses of $CH_4$ from air extracted from the horizontal ice core in Taylor Glacier, Antarctica, covering the Younger Dryas-Pre-Boreal transition constrain that the

contribution from old, $^{14}C$-free, carbon—which includes methane hydrates, permafrost and methane trapped under ice—to the rapid $CH_4$ rise at 11.5 kyrs BP was <19%[40]. This further supports the idea that the abrupt rises in both greenhouse gases ($CO_2$ and $CH_4$) at the end of the last deglaciation might not have been caused by permafrost thawing and degradation alone. Nonetheless, our results of the flooded shelf and hinterland permafrost thawing inherently imply the rapid subsequent initiation of wetland formation in the large Amur and other Siberian catchments, which established significant boreal 'young carbon' $CH_4$ sources in lockstep with our observed activation of previously frozen, inert old carbon from terrestrial reservoirs.

Our findings provide the first direct evidence for rapid mobilization of old permafrost-derived carbon in boreal to subarctic East Asia during the last glacial termination, prior to the onset of

the Pre-Boreal. This substantial activation of pre-aged carbon (5 to 10 kyrs at the time of deposition) supports modelling studies published in recent years, which considered this process as a possible cause for abrupt $CO_2$ releases. High accumulation rates of pre-aged terrestrial biomarkers at times of rapid sea level rise (melt water pulse 1A and 1B) suggest that shelf erosion was the dominant process for carbon mobilization. However, fluvial export of old carbon from degrading permafrost in the Amur hinterland represents another important process for mobilization, particularly during times of high river discharge (~10 kyrs BP).

The extrapolated carbon cycle changes led to simulated $CO_2$ changes which are about a quarter (16.5 kyrs BP) or a half (14.6 and 11.5 kyrs BP) of the size of the three individual peaks found in the WAIS Divide ice core, but on the long-term to less than 5% of the deglacial rise in atmospheric $CO_2$ of ~90 ppm[12]. Moreover, only little (<10 ‰) to the residual in the atmospheric $\Delta^{14}C$ decline across Termination I that is unexplained by changes in the $^{14}C$ production rate are according to our results related to this permafrost carbon release. Altogether, this implies that deglacial changes in atmospheric $CO_2$ and $\Delta^{14}C$, while largely controlled by oceanic processes, were additionally impacted by degrading permafrost, potentially partly accounting for the abrupt $CO_2$ rises at 14.6 and 11.5 kyrs BP that so far have remained difficult to explain[3,12]. Further investigations of permafrost-carbon mobilization from locations bordering the permafrost domain that have undergone significant deglacial changes are needed to improve the quantification and constrain the possible age ranges of the mobilized carbon as well as their potential climate feedback.

## Methods

**Study area and core locations.** The circulation in the Okhotsk Sea is dominated by the Okhotsk Gyre and includes the southward-flowing East Sakhalin Current (ESC), which transports surface and deep waters from the northern shelves to the Kuril Basin[41]. During the sea-ice season in fall and winter, Dense Shelf Water (DSW) is formed and flows south along the Sakhalin margin, transporting high concentrations of organic matter, lithogenic particles and suspended matter that are entrained by vigorous tidal mixing on the northwestern shallow continental shelf into a highly turbid water layer[10,42]. Combined with discharge from the Amur River these materials rapidly accumulate along the East Sakhalin margin[10,42], making this location the primary depositional site for terrigenous sediments supplied to the Okhotsk Sea.

The two cores used in this study were retrieved from the northeast Sakhalin margin within the framework of the German-Russian KOMEX I and KOMEX-SONNE projects in 1998 and 2004, respectively. Gravity core LV28-4-4 (51°08.475′N, 145°18.582′E, 9.3 m recovery) was collected from 674 m water depth during expedition LV28 with R/V *Akademik Lavrentiev*[43] and piston core SO178-13-6 (52°43.881′N, 144°42.647′E, 23.7 m recovery) was collected from 713 m water depth during the expedition SO178 with R/V *Sonne*[44]. The two cores feature similar lithofacies, consisting mainly of silty clays with sand and occasional larger dropstones derived from sea-ice transported terrigenous matter.

As both cores were retrieved from a relatively dynamic sedimentary setting, we took care to select sites that were both (1) representative of the depositional environment targeted (river discharge and DSW transport), and (2) undisturbed by secondary processes, such as sediment redistribution, mass wasting, chaotic facies, etc., as evidenced through prior extensive seismic survey works. Both cores were retrieved from areas surveyed with a purpose-built high-resolution sub-bottom profiling system (SES 2000 DS), with decimeter-resolution to sediment penetration depths of about 30 m below sediment surface. No disturbances, but flat, continuous, undisturbed sedimentary facies across several hundred metres or even kilometres were recorded on both core locations. For the SO178-13-6 location, in addition the shipboard PARASOUND sub-bottom profiler system was extensively used to confirm the absence of slumping, turbidites, mass wasting or other processes that might obfuscate or invalidate our assumptions about our sedimentary recording system of southward transport processes along the Sakhalin margin. In summary, both locations together represent an optimal approach for recording the depositional history of terrigenous transport from the Amur and Siberian hinterland.

**Sediment chronology.** Core SO178-13-6 covers the last 17 kyrs. The chronology was established through an AMS $^{14}C$-anchored stratigraphic framework based on XRF scanning and core-to-core correlations for the Okhotsk Sea during the last deglaciation[45]. The chronostratigraphic age models for both cores used here have been published in detail previously and are used here without changes[46]. LV28-4-4

covers approximately the last 16 kyrs BP. Age control was obtained through 12 AMS $^{14}C$ dates on *G. bulloides* and *N. pachyderma* sinistral, supplemented by eight AMS $^{14}C$ dates on mollusks/gastropods. All AMS $^{14}C$ ages were calibrated with a regional reservoir correction of R 500 ± 100 yr and the MARINE09 calibration curve[47]. The routine CLAMS[48] written in R was used to find a best fit through the $2\sigma$ ranges of all age control points, resulting in a smooth spline fit (0.3 smoothing factor) with a final run with 10,000 iterations in the CLAMS routine[46]. The age models revealed particularly high sediment accumulation rates during the deglacial period (8–18 kyrs BP) for core SO178-13-6, whereas maxima in sediment accumulation are reached in the Holocene in core LV28-4-4. The cores were sampled with varying resolution between 5 and 40 cm for down core analyses. Large samples of 50–100 g (dry weight) sediment for compound-specific radiocarbon analysis (CSRA) were taken from 4 (core LV28-4-4) and 6 (core SO178-13-6) depth horizons. Note that the lowermost sample of LV28-4-4 at depth 926–928 cm is below the last AMS $^{14}C$ date used for the age model. The depositional age was linearly interpolated from there.

Chlorophycean freshwater algae counts were carried out on 32 pollen samples of core LV28-4-4 (*Pediastrum* spp.). Sample preparation and counts were reported in detail in previous studies[46].

**Analytical methods.** We present accumulation rates and concentrations (Supplementary Fig. 2) of two groups of terrigenous biomarkers, i.e., long-chain *n*-alkanes and branched glycerol dialkyl glycerol tetraethers (brGDGTs). Long-chain *n*-alkanes like long-chain *n*-alkanoic acids are primarily derived from leaf-waxes of higher plants, but their concentrations can more reliably be determined by gas chromatography, as the analytical procedures are more robust. To assure that their concentration records are not affected by petrogenic input and reliably reflect degrading permaforst, we compared them with concentration records of brGDGTs derived from soil bacteria. For down-core biomarker analysis total lipids were extracted from freeze-dried, homogenized sediment (2–5 g) using a three-step ultrasonic extraction with (i) dichloromethane, (ii) dichloromethane:methanol 1:1 (v:v) and (iii) methanol. Total lipid extracts were hydrolysed with 0.1 M potassium hydroxide (KOH) in methanol:water 9:1 (v:v) at 80 °C for 2 h in order to separate *n*-alkanoic acids (from here on called fatty acids (FAs)) from neutral lipids (NLs). Neutral compounds were extracted with *n*-hexane. Subsequently, the pH was adjusted to 1 by adding 37% HCl and FAs were extracted with dichloromethane. NLs were further split into three subfractions (apolar compounds, aldehydes and ketones, polar compounds) by silica gel chromatography. The *n*-alkane concentrations of core LV-28-4 were determined using a HP 5890 GC and of core SO178-13-6 using an Agilent 7890A GC. Both GCs were equipped with an Agilent J&W DB-5ms column and a flame ionization detector (FID). Each compound was identified based on retention time and comparison with an *n*-alkane standard. Quantification was achieved using an internal standard (squalane) added prior to extraction. brGDGTs were analysed with minor modifications according to ref. [49]. Briefly, the polar fraction was filtered through 0.45 µm PTFE syringe filters and dissolved in hexane:isopropanol 99:1 (v:v). Samples were analysed using an Agilent 1200 series HPLC coupled to an Agilent 6120 single quadrupole MS via an atmospheric pressure chemical ionization interface (APCI). Chromatographic separation was achieved by normal-phase chromatography using a Prevail Cyano column (Grace, 3 µm, 150 mm × 2.1 mm) maintained at 30 °C. brGDGTs were identified using selective ion monitoring[50] and quantification was achieved using an internal standard ($C_{46}$-GDGT) added prior to extraction.

Samples for CSRA were extracted from freeze-dried and homogenized sediment for 48 h using a Soxhlet with a mixture of dichloromethane:methanol 9:1 (v:v). The total lipid extract was hydrolysed as described above and the retrieved FAs were derivatized to fatty acid methyl esters (FAMEs) by adding HCl and methanol of known $\Delta^{14}C$ reacting in a nitrogen atmosphere at 80 °C overnight. FAMEs were extracted from the methylated solution with *n*-hexane and subsequently separated from polar compounds with silica gel chromatography. In preparation for purification of individual FAMES, branched and unsaturated FAMEs were removed from the FAME-fraction extracted from core LV28-4-4 using urea adduction and a column of silica gel coated with silver nitrate, respectively. For samples of core SO178-13-6 it was sufficient to clean samples with silica gel chromatography as urea adduction was not necessary. For CSRA the *n*-$C_{26:0}$ and *n*-$C_{28:0}$ alkanoic acids were purified using preparative capillary gas chromatography (PC-GC)[51] performed on an Agilent HP6890N GC with a Gerstel Cooled Injection System (CIS) connected to a Gerstel preparative fraction collector[52]. The GC was equipped with a Restek Rtx-XLB fused silica capillary column (30 m, 0.53 mm diameter, 0.5 µm film thickness). All samples were injected stepwise with 5 µL per injection. Purified compounds were transferred to pre-combusted quartz glass tubes and 150 µg pre-combusted copper (II)-oxide was added as oxidizing agent. Quartz tubes were evacuated ($10^{-5}$ mbar) on a vacuum line and flame-sealed with a hydrogen/oxygen torch. The sealed tubes were combusted at 950 °C for 4 h. The resulting $CO_2$ was stripped from water and quantified.

**Compound-specific radiocarbon analysis.** The isotopic ratio ($^{14}C/^{12}C$) of the $CO_2$ samples derived from the individual *n*-alkanoic acids was determined by Accelerator Mass Spectrometry (AMS). The measurements were carried out on the MICADAS-system equipped with a gas-ion source[53,54] at the Laboratory of Ion Beam Physics, ETH Zürich. Samples were normalized and background subtracted

using Oxalic Acid II (NIST SRM 4990C) and radiocarbon free $CO_2$ gas. AMS results are reported as fraction modern carbon ($F^{14}C$; Supplementary Table 1), conventional radiocarbon ages ($^{14}C$ ages given in $^{14}C$ yr BP), and $\Delta^{14}C$[55].

**$^{14}C$ blank assessment and corrections.** In order to report accurate radiocarbon dates of terrigenous biomarker compounds, a correction for extraneous carbon of unknown composition that is introduced to the sample during processing is required (procedure blank). Possible contamination during sample preparation prior to AMS analysis include carbon added through column bleed and carry-over as well as atmospheric carbon during vacuum line handling and combustion due to leakages. Therefore, every measured $F^{14}C$ of a processed biomarker sample ($F^{14}C_{sample}$) is a composite of the true $F^{14}C$ of the biomarker ($F^{14}C_{true}$) and the $F^{14}C$ of the blank ($F^{14}C_{blank}$). Blank correction of AMS measured $F^{14}C_{sample}$ requires the determination of $F^{14}C_{blank}$ and the size of the blank ($m_{blank}$)[56]. Assuming constant $F^{14}C_{blank}$ and a constant mass of the blank ($m_{blank}$), there is an inverse linear relationship between the $F^{14}C_{sample}$, and the sample size[57] ($m_{sample}$; Eq. 1):

$$F^{14}C_{sample} = a \times \left(\frac{1}{m_{sample}}\right) + F^{14}C_{true} \tag{1}$$

where the slope ($a$) is defined as:

$$a = m_{blank}\left(F^{14}C_{blank} - F^{14}C_{true}\right) \tag{2}$$

Given these relationships, $m_{blank}$ and $F^{14}C_{blank}$ can be derived from the intersections of two linear regression functions obtained from processing several, ideally different sized, samples of at least two different materials (1,2) with known (different) $F^{14}C_{true}$ (Supplementary Fig. 3) as:

$$m_{blank} = \frac{a_1 - a_2}{\left(F^{14}C_{true1} - F^{14}C_{true2}\right)} \tag{3}$$

In order to assess the procedure blank, we processed n-hexadecanoic acid (n-$C_{16:0}$) from apple peel collected in 2013 (modern $F^{14}C_{true}$) and n-triacontanoic acid (n-$C_{30:0}$; Sigma, Prod. No. T3527-100MG, LOT 018K3760, fossil $F^{14}C_{true}$; Supplementary Table 2). $F^{14}C_{true}$ of n-$C_{30:0}$ has been reported[58]. The $F^{14}C_{true}$ of the apple peel was obtained from AMS-analyses of bulk apple peel (3 samples, graphitized and analysed at ETH Zürich) assuming that the $F^{14}C_{true}$ of n-$C_{16:0}$ equals the $F^{14}C$ of bulk apple peel. The $F^{14}C_{true}$ and $F^{14}C_{sample}$ of apple peel and n-$C_{30:0}$ as well as the respective sample sizes are given in the Supplementary Table 2. $a_1$ and $a_2$ (Eq. 3) were assessed from linear regression (Supplementary Fig. 3). For the blank assessment, it has to be acknowledged that the processed n-$C_{16:0}$ and n-$C_{30:0}$ were methylated for GC analysis. This means that $F^{14}C_{sample}$ is affected by the $F^{14}C$ of the added methyl-group ($F^{14}C_{methyl}$) while the $F^{14}C_{true}$ (unprocessed equivalents) is not. Hence, when determining the $m_{blank}$ and $F^{14}C_{blank}$ as discussed above and shown in Supplementary Fig. 3, the methyl group of the processed n-$C_{16:0}$ and n-$C_{30:0}$ FAMEs affects the slope of the regression lines. As a result, this would count towards the unknown blank. In order to remove the contributions of the methyl group from the blank assessment, the $F^{14}C_{true}$ values the unprocessed n-$C_{16:0}$ and n-$C_{30:0}$ FAs would have if they were methylated has to be calculated. This was achieved by combining the $F^{14}C_{true}$ of the bulk apple peel and the n-$C_{30:0}$ FA with the $F^{14}C_{methyl}$ through isotopic mass balance.

The uncertainties of $F^{14}C_{blank}$ and $m_{blank}$ ($\sigma F^{14}C$ and $\sigma_m$) can be inferred from the regression coefficient ($R^2$)[57] as $R^2$ indicates the certainty $F^{14}C$ and $m$ are predicted with by the regression line. Under this assumption $\sigma F^{14}C$ and $\sigma_m$ can be obtained from:

$$\sigma F^{14}C = F^{14}C_{blank} \times \left(1 - R^2\right) \tag{4}$$

$$\sigma_m = m_{blank} \times \left(1 - R^2\right) \tag{5}$$

The $R^2$ of the n-$C_{16:0}$ regression was considered only, as the n-$C_{30:0}$ regression was based on $n = 2$ (Supplementary Fig. 3). Blank correction (of the CSRA results from the biomarker samples) and error propagation was performed after[56]. The obtained $F^{14}C$ values were corrected for $F^{14}C_{methyl}$ by isotopic mass balance.

**Calculation of terrigenous biomarker ages at deposition.** We use the measured $\Delta^{14}C$ signature of our terrigenous biomarkers to calculate the age at deposition as follows: any carbon of terrestrial plants has its origin in atmospheric $CO_2$ and therefore will incorporate carbon with the isotopic $^{14}C$ signature of the atmosphere at the time of photosynthetic production. Using the atmospheric $\Delta^{14}C$ record of IntCal13, we can therefore calculate the radioactive decay of $^{14}C$ (using the decay constant $\lambda = 8267^{-1}$ yr$^{-1}$) of any sample as function of age and can derive the $\Delta^{14}C$ signature this sample should have during the time of the measurement nowadays, in our case in the year 2013 and 2014 for the cores LV28-4-4 and SO178-13-6, respectively. By using this IntCAL13-based age calculation as look-up table, we are able to determine from our measured

$\Delta^{14}C$ in which year the biomarkers have been photosynthetically fixed. The age at deposition can then be derived from the difference between the time of photosynthetic production and the deposition age (Table 1). The uncertainty of the age at deposition is estimated from the uncertainty in the $\Delta^{14}C$ measurement, to which an additional $1\sigma$-uncertainty in age of 10 years is added, which accounts for the uncertainty of $\Delta^{14}C$ in the IntCAL13 record. This uncertainty offset fully accounts for any potential error propagation for the range of our $\Delta^{14}C$ data (from $-358$‰ to $-932$‰).

**Estimated impact on atmospheric $CO_2$ and $\Delta^{14}C$.** Using the well-known carbon cycle box model BICYCLE[59], we estimated how much the coastal erosion throughout the Arctic Ocean might have impacted both atmospheric $CO_2$ and $\Delta^{14}C$. For this aim, we perturbed simulation results covering the last glacial cycle. In our control run (using the atmosphere-ocean subsystem including carbonate compensation to simulated ocean-sediment interactions) the carbon content of the terrestrial biosphere was kept constant. In our additional simulations it was prescribed to release $CO_2$ from thawing permafrost across the last deglaciation (18–10.8 kyrs BP). We are here mainly interested in the overall effect, therefore prescribed $CO_2$ release rates from permafrost thawing were either kept constant over the last deglaciation, or restricted to three peaks potentially connected with rapid sea level rise. In the latter scenario carbon is released to the atmosphere in three 200-year time-windows starting at 16.5, 14.6 and 11.5 kyrs BP. The amount of released carbon in the three peaks is 20%, 40% and 40% of the total released carbon of 85 PgC, respectively, to approximately account for the amplitude differences in the maxima in our MAR record. These carbon releases are assigned with a $^{14}C$ signature that has been constructed from a pre-depositional age of either 5 or 10 kyrs derived from the CSRA analyses of the two sediment cores discussed in this study (Supplementary Fig. 1b). Alternatively, we added another scenario in which $^{14}C$-free carbon is entering the atmosphere as proposed before[3], which would imply a pre-depositional aging of the carbon of up to 35 kyrs resulting in approximately a doubling of the simulated $\Delta^{14}C$ peak (e.g. $-12$‰ for the 14.6 kyrs BP event, Fig. 5b). However, the Tahiti-based $\Delta^{14}C$ anomaly in focus in ref. [3] is even not entirely met, when the background trend of $-0.1$‰ per year, potentially caused by marine processes and changes in the $^{14}C$ production rate, is added to our simulation results (Fig. 5b). For a strict process separation $^{14}C$ production rate was kept constant here, at 25% higher than modern times in order to obtain a simulated atmospheric $\Delta^{14}C$ level of 400‰, similar to what is found in the IntCAL13 $^{14}C$ stack around 18 kyrs BP[11].

Previous simulations with the same model[60,61] show a $^{14}C$ production-corrected residual of atmospheric $\Delta^{14}C$, which needs to be explained by carbon cycle changes. This residual in atmospheric $\Delta^{14}C$, using the $^{14}C$ production rate estimate based on the geomagnetic field reconstruction[62] GLOPIS-75, has a large uncertainty band of about $\pm 80$‰ around a mean decline of 100‰ across the last deglaciation (18–11 kyrs BP) (Fig. 4b). Note that in a more recent study using a similar carbon cycle box model[63] this $^{14}C$ production corrected $\Delta^{14}C$ residual is by about a factor of two larger than these results obtained with the BICYCLE model.

To put our carbon cycle simulation results into context with other models[64,65], we compare their model response to an external perturbation. In detail, the fraction of carbon injected into the atmosphere that stays there, the so-called airborne fraction, is compared for a pulse of 100 PgC injected to the atmosphere within 1 year for pre-industrial (PI) climate conditions (Supplementary Fig. 1a). This quantity is a function of time and evaluates our simulated anomalies in atmospheric $CO_2$. The results of a model intercomparison project (MIP)[64] are restricted to 1 kyr. In order to evaluate our results for the whole period of the last deglaciation, we extrapolate this MIP-based airborne fraction with exponential equations obtained from regression analysis of Earth system model simulation results[65]. While this approach provides only an approximation of the evolution of the mean value, we keep the relative width of the $2\sigma$ uncertainty band, based on MIP for 1 kyr, to show the likely uncertainty range or model spread for longer time periods.

## Data availability

Data obtained in this study are deposited in Pangaea (https://doi.pangaea.de/10.1594/PANGAEA.890865).

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

## Acknowledgements

We acknowledge the professional support of masters and crew of R/V Akademik M.A. Lavrentiev and R/V Sonne on expeditions LV28 and SO178. We thank Ralph Kreutz and Hendrik Grotheer for laboratory support. We thank Jens Strauss for valuable comments on the manuscript. G.M. acknowledges funding from the Helmholtz Association (grant no. VH-NG-202 also supporting M.W., and funds in the W2/W3 program supporting V. M. and J.H.). L.L.-J. and R.T. acknowledge support from the Helmholtz REKLIM Initiative and BMBF grant no. 03F0704A 'Sino-German Pacific-Arctic Experiment (SIGEPAX)'. U.K. acknowledges support from the VILLUM Foundation (grant no. 10100). This work contributes to PALMOD, the German Paleomodeling Research Project funded by BMBF.

## Author contributions

G.M. designed the study. R.T. and L.L.-J. performed field work and sampling. M.W., W.D. and J.H. carried out biomarker analyses, purification of long-chain *n*-alkanoic acids, and preparation for compound-specific radiocarbon analysis. C.M. and L.W. performed AMS [14]C analysis on the MICADAS system and helped with [14]C data processing. V.M. was responsible for [14]C blank correction. P.K. performed the carbon cycle modelling and calculated the pre-depositional ages. U.K. provided freshwater algae counts. M.W. and G.M. wrote the manuscript with input from L.L.-J., V.M., W.D., P.K. and L.W. All authors discussed the results and commented on the manuscript at different stages.

## Additional information

**Competing interests:** The authors declare no competing interests.

