## [Peer Review File · Nature Communications]

Reviewers' comments:

Reviewer #1 (Remarks to the Author):

Review of: Deglacial mobilization of pre-aged terrestrial carbon from thawing permafrost.

Winterfeld et al. 2017

Overall evaluation:

The manuscript presents direct evidence for the existence of a permafrost carbon pool during the last glacial maximum based on data gathered from sediments cores offshore of the Amur River basin. The cores contain organic molecules of terrestrial origin that were thousands of years old at the time of deposition in the ocean. This indicates the organic matter was derived from permafrost soils and was transported into the ocean via physical erosion associated with sea-level rise and deglacial thawing of permafrost soils. Back-of-the-envelope estimates of carbon release via erosion, combined with carbon cycle modeling show that the carbon released could explain spikes in atmospheric CO₂ concentration associated with the melt-water pulses.

The paper has two halves: (1) an extensive analysis of the Okhotsk Sea sediment cores, which presents excellent direct evidence of the existence of a glacial permafrost carbon pool. And (2) a modelling exercise linking abrupt sea-level-rise to the release of permafrost carbon a spikes in atmospheric CO₂ concentration. The first part of the paper is original, ground-breaking and deserving of publication in a high profile journal. The modelling component of the paper has some flaws. I recommend publication with revisions.

General Comments:

(1) The modelling component of the study tries to establish physical erosion of Yedoma deposits as the principle mechanism of permafrost carbon mobilization during the melt-water pulses. The much of the modern Yedoma deposits do boarder the Eastern arctic sea (Strauss et al. 2013), and up to 80% of the flooded shelf is estimated to have been part of the Yedoma region during the LGM (Walter et al. 2007). However, the authors appear to have overestimated the amount of carbon likely held in these flooded Yedoma deposits, and do not establish how that coastal erosion could have mobilized such a large fraction of this organic matter.

The modern Yedoma deposits are estimated to contain 181 PgC (Hugelius et al. 2014). If 80% of the 1.9 million km² of Eastern arctic shelf was Yedoma region in the last LGM and if the region had the same carbon density of the modern deposits, then the area would contain ~200PgC. This estimate is smaller than the authors estimate (285 PgC) but the more important issue is how much of the carbon would be mobilized when the region flooded?

The authors assume that all of the carbon in the flooded region was mobilized, however they do not provide evidence to support this assumption. Certainly some of the carbon would be mobilized through erosion of sea-cliffs but how deeply where cliffs eroded? Yedoma deposits are characteristically 10s of meters thick assuming that the coast was eroded by that depth of soil over such a large area seems extraordinary. Presumably much of the cost would have been relatively passively flooded resulting in the wide-spread subsea permafrost seen in the region today (Shakhova et al. 2010). That is, much of the LGM permafrost carbon may still be frozen in the permafrost of the East Siberian Arctic shelf, below sea-level but kept frozen by ocean temperatures below 0°C.

I agree that the LGM permafrost carbon pool could have played a key role in the spikes in atmospheric CO₂ seen during the melt-water pulses. However taking into account the response of the inland permafrost carbon pool to the abrupt warming during theses intervals also seems crucial.

I recommend that either the authors present evidence to justify release of 170 PgC via coastal erosion or significantly rework the modelling component of the paper. I believe the first part of the paper focusing on the sediment cores from the Okhotsk Sea is compelling enough to be the sole focus of the paper.

Specific comments:

Units: Both kyr and ka are used. I prefer ka but you may want to consult Nature's style guidelines.

Line 62: Rework this sentence to better orient the reader. Most will not know that the Amur river forms the much of the boarder between China and Russia.

Line 91: I recommend writing out Pre-boreal everywhere instead of using an abbreviation.

Line 98: Change "rather" to "relatively"

Line 105 to 106: Weaken this statement, i.e. 'must have' to 'likely have'

Line 198: Change "rather" to "relatively"

Line 196: Change "Arctic region" to "permafrost region". Most of the LGM permafrost was well outside the Arctic. The Laurentide ice sheet extended below 40°N, and the permafrost region extended well south of the ice sheet edge.

Line 221: Write out "Ref. 3"

Line 278 to 279: This sentence does not make sense. Total loss and rate of loss are being conflated.

Line 308: Change "the last couple of years" to "recent years"

Line 494: Change "While in" to "In"

Line 497: Change comma to period. Re-write next sentence to "While in our additional simulations it was prescribed"

References:

Hugelius, G., et al., 2014: Estimated stocks of circumpolar permafrost carbon with quantified uncertainty ranges and identi_ed data gaps. *Biogeosciences*, 11 (23), 6573–6593.

Shakhova, N., I. Semiletov, A. Salyuk, V. Yusupov, D. Kosmach, and O. Gustafsson, 2010: Extensive methane venting to the atmosphere from sediments of the East Siberian Arctic Shelf. *Science*, 327, 1246–1250.

Strauss, J., L. Schirrmeister, G. Grosse, S. Wetterich, M. Ulrich, U. Herzschuh, and H.-W. Hubberten, 2013: The deep permafrost carbon pool of the Yedoma region in Siberia and Alaska. *Geophysical Research Letters*, 40 (23), 6165–6170.

Walter, K., M. Edwards, G. Grosse, S. Zimov, and F. Chapin, 2007: Thermokarst lakes as a source of atmospheric ch₄ during the last deglaciation. *Science*, 318 (5850), 633–636.

Reviewer #2 (Remarks to the Author):

Winterfeld et al. used two sediment cores collected from the east of the Sakhalin Island and used these two cores to discuss the history of permafrost thawing and exporting from the Amur River drainage basin over the last ~17Ka.

The authors measured carbon values, specific biomarker concentrations and radiocarbon values of fatty acids in selective layers. The authors concluded that there were three major periods of permafrost thawing based on biomarker accumulation rates and may have largely affect atmospheric CO₂ levels. The authors further applied a model and tried to apply the model in explaining atmospheric CO₂ variation and its linkage with permafrost thawing in the Arctic. This manuscript presented two sediment cores covering such long time scales and these two cores are important in understanding the Arctic permafrost dynamics over time. This study also presented data showing the significant differences of carbon dynamics in the coastal Arctic Ocean in late Pleistocene and how the carbon dynamics vary over time. This study also used the state-of-the-art technique – compound specific radiocarbon analysis in understanding how the sources of sedimentary carbon may vary over time. I find the topic is quite interesting as permafrost thawing is a hot topic now since it is sensitive to climate change under current circumstances.

However, I also find this manuscript has many issues. For example, many discussion and conclusions are not well supported by either the data or the way of data interpretation using proxies. I find it hard to separate the novelty and validity of this study and how the conclusions could contribute to our current knowledge on the last glacial. Tesi et al., 2016 (Nature Comm) presented a case study suggesting massive remobilization of permafrost carbon during post-glacial warming. Although there are some issues involved in that manuscript, based on my opinion, that manuscript has answered most of the major conclusions that have been drawn in this study. Therefore, it is pretty hard to see the major new conclusions in this study. In addition, some of the conclusions are rather weak. Furthermore, some of the issues are non-fixable, such as the bias of the sampling locations. Therefore, based on my humble opinion, I think this manuscript does not meet the requirement of Nature Communications, but could be suitable for a more specific journal.

Major issues:

1) The sampling locations and how the sedimentary records are linked with permafrost thawing are susceptible. Based on figure 1, the sampling locations are on the east of the Sakhalin Island and are pretty far away from the current mouth of the Amur River. Currently, the Amur River drains into the north side of the Strait of Tartary west of the Sakhalin Island. To what I know, it may have minimal effect to where the samples were taken. However, during last glacial period, the Strait of Tartary was above water and the river mouth was inland. Therefore, the river mouth should be at a different location. The Amur River most likely drains to a location north of the current one.

Based on figure 1, the sampling location, especially SO178-13-6 was close to the coast and had a shallow water depth back in last glacial period. However, with sea level rise, the water depth went deeper. It is also worth note that the water depths of these two cores are different, with core LV28-4 much deeper. Therefore, it is very likely that core SO178-13-6 was largely affected by coastal processes since it is closer to the coast. When the water depth went deeper, it definitely has less effect from coastal erosion. However, since these two sampling locations are pretty far away from the river mouth, and may not have direct linkage with the river discharge, I highly doubt the sedimentary record is representative of what happened in the drainage basin. Instead, it may represent what happened on the Sakhalin Island since it is closer. Furthermore, I don't understand why authors picked two cores and only did minimal work on the second core. The authors could have done more CSRA on one core to improve the resolution of radiocarbon analysis.

When I read through the manuscript, I see one major focal of the manuscript is to distinguish

carbon sources of coastal erosion and hinterland inputs through river transport. However, based on my concerns, I highly doubt the validity of the discussion.

2) Based on my knowledge, I don't think the modeling part makes sense at all. The model was based on the assumption that ~60% of the permafrost deposited on the Arctic shelf could have been released during last glacial time because the sea level raised by 60% from ~ -120m to ~ -50m. However, I don't understand the logic here since whenever sea level rises, especially when rise rapidly, this ~60% of the permafrost deposition was supposed to be submerged under water and be buried quickly by later deposition from terrestrial sources. I got further confused when authors assign this assumed ~170PgC release into three time periods based on peak periods of biomarker accumulation. I was totally lost when authors have the whole modeling discussion based on such assumptions. In addition, in terms of modeling and real data, it should always been modeling based on real data, other than data interpretation based on modeling. In this case, the number of ~170PgC release was calculated based on a simple model and this number has been widely discussed and applied through the majority of the discussion. Therefore, I don't think the discussion is valid at all.

Minor comments:

Line 54: Does not follow the logic of the previous sentence and is a little bit odd here.

Line 55: The general audience may not know the exported material includes particulate and dissolved. Please make it clear.

Lines 56-59: This is not clear on how radiocarbon ages of terrestrial carbon can help answer carbon-climate feedback.

Line 61: Don't think it is possible considering the old ages and mixing of variable carbon ages during export.

Line 64: change Amur to Amur River

Lines 68-70: In this case, the sediment core, even terrestrial biomarker radiocarbon age doesn't necessarily represent terrestrial material discharged from the Amur River.

Line 70: As I stated in the major comments, the sampling cores are one of my biggest concern here as back in time when sea level was lower the Straight of Tartary was above sea level and Amur River drains across Sakhalin and may largely affect the sampling locations in this study. However, during deglacial time and the Holocene when the sea level was higher with the formation of the Straight of Tartary, the Amur river drains into the Straight of Tartary directly and doesn't affect the sampling locations at all. For example, if you check on the current world river map, the Amur River drains into the north part of the Straight of Tartary. This may also explain the high sedimentation rates and OCAR at some time periods. Therefore, the OCAR may not represent how climate shifts affect carbon exports, but more representative of drainage basin size that affects these sampling locations.

Lines 70-71: These two sediment cores are far away from each other and apparently receive different materials with sorting of material during transport. There has been study that hydrodynamic sorting has an effect on the age of carbon.

Line 73: Could also be from the erosion of petrogenic sources?

Line 74: BrGDGT is only representative of soil erosion, not fresh material inputs and is highly biased by soil development.

Line 76: what's the purpose of analyzing different biomarkers for concentrations and radiocarbon age? Could have been much more helpful if fatty acid concentrations are reported as well.

Lines 87-88: How is this compared with the MAR and OCAR. I suspect they all have the same trend and all the trends are ultimately controlled by the sedimentation rates, which is linked with material inputs. Further, material inputs are mainly affected by the offshore distance and riverine inputs.

Lines 95-96: it doesn't make sense here as B/A and PB are the two warmer periods before and after YD. Without evidence showing coastal erosion, it could also be the release of carbon from the drainage basin. Even so, coastal erosion and drainage basin inputs most likely work together.

Lines 96-97: The abrupt increase in biomarker accumulation, to me, is due to much higher sedimentation rates. I suspect biomarker concentrations don't change much through that time period. In addition, vegetation change at warming period may also contribute more HMW alkanes to the coast.

Line 101: During extreme events when sedimentation rates are much higher, the oxygen exposure time is shorter and the degradation is inhibited. I would like to see any degradation index through time to support the supposition of OC degradation.

Line 101: Why not transport to deep sea? What is the ice process in this region? Will that also affect the burial of terrestrial material?

Line 114: the resolution of fig 2f is rather low, not many data points corresponding to the peaks in 2c/d. It is quite weak to make this statement.

Lines 117-120: Again, poor sampling for fig 2f, you cannot make such statement. The biomarker in fig 2f is derived from aquatic sources, while 2c/d are biomarkers for terrestrial sources in general, specifically soils and plants. They don't have to match with each other. But as I mentioned earlier, the evidence for coastal erosion is not well supported. Do you have any evidence showing where the Amur River has drain into the sea? Also, discharge and biomarker data are from two different cores which are quite far, so it is reasonable to have some timing difference.

Line 120: I doubt the application of algae biomarker in reconstructing discharge. To me, it is more representative of aquatic algae growth. This is a totally different term to discharge and they may vary independently.

Line 126: What's the purpose of analyzing 2 cores with such far distance? Instead, by doing one core, you could increase the resolution of radiocarbon samples. I only see five and three on the figure3C. Where are the two other samples?

Lines 128-129: I don't think your data is showing this way. There are only eight data points, and only 2 after PB. For these 2 data points, one is comparable in age to the oldest pre PB, the other one is slightly lower than the youngest age pre PB. I understand CSRA requires a lot of work, but even so, such low sampling resolution could not support this statement.

Line 134: What is the yedoma distribution in this region?

Line 135: Too many uses of vast in the MS.

Lines 145-146: The only evidence showing this is material from hinterland is through algae biomarker, which is suggestive of freshwater inputs. However, I think as a major conclusion of this manuscript, the discussion on terrestrial inputs through riverine transport is a bit of weak. You should find stronger evidence suggesting this is through riverine transport.

Line 149: I think the results are not well discussed in terms of coastal erosion and hinterland inputs. It definitely makes sense that when the sea level was low, the sampling locations were way closer to the coast and therefore receive massive material from coastal erosion. While when sea level was higher, the sampling locations were away from the coast, and therefore were less affected by coastal erosion. However, I am surprised to see no discussion on water depth and its effect on bulk and biomarker signals.

Lines 151-156: Don't quite understand this part. Do you mean the material eroded now and 10 Ka ago is the same when considering the modern core top material is 18Ka, while the material 10Ka ago was 8 Ka when deposited? Mathematically it is correct, however, in reality, I doubt the truth of this statement. The permafrost landscape should have changed significantly with time considering the sea level, humidity, vegetation growth and human disturbance, etc. Thus such statement based just on carbon age is weak.

Lines 158-159: Don't see fatty acid data support this statement.

Line 161: Not all Arctic, please be more specific.

Line 163: It should be a range based on Figure 3C.

Line 166: The authors used a lot of modern analogs. However, in this case, the modern analog of coastal erosion only has carbon age of ~3-5 ka, much younger than the 10ka in this statement. I don't really believe it is through coastal erosion. Warming induced sea level rise could also stimulate export of terrestrial material through river transport. However, it is important to evaluate how largely the river affects this region first.

Line 175: Paq is an empirical index and is affected by many factors, such as degradation, sources. I don't think it is the ideal proxy to be included in this case. Paq could also be affected by marine inputs, that is barely discussed in this manuscript. Terrestrial inputs, including nutrients, could stimulate marine primary production as well.

Line 188: Throughout the manuscript, there is no proper discussion of permafrost carbon

degradation. Although many cases have been published on modern systems, the statement of permafrost degradation is not a conclusion that can be reached in this study.

Lines 191-193: add references indicating the glacial retreat in these areas.

Lines 197-202: Is this only on Pleistocene time scale or from Pleistocene to the present?

Line 200: height to thickness.

Line 210: Don't understand how this carbon could possibly be fully oxidized into CO₂ since it has been buried under sea during sea level rise with extra material accumulating on top of the yedoma since once underwater there are deposits of new material from terrestrial sources or marine sources. This carbon pool is likely permanently buried with minimal interaction to the atmosphere. However, I am surprised to see the whole model is based on this assumption, which to me is susceptible. See my major comment for details. To me, I think the majority role of permafrost thawing on atmospheric CO₂ level is through degradation of DOC during inland and riverine transport and soil respiration, however, the POC is less likely the main driver of atmospheric CO₂ levels.

Line 213: I don't think it is doable this way by forcing 170PgC into these three events, especially when considering this number is generated using a simple box model.

Line 221: Again, I am not sure if it is right to set the same time period of 200 for all three events. Check on the grammar of the sentence as well.

Line 222: I thought 14.6 Ka pins the start of carbon release in your case? I don't understand why 16.5 is included here. I can see the 16.5Ka peak is much smaller than the 11.5 Ka, so I don't really understand why the release rate is even higher for the 16.5 Ka.

Lines 222-225: 1) I don't understand how this number is generated; 2) don't understand the mean of release here. Do you mean the released carbon goes to the sedimentary pool or the atmosphere? I don't think it is right to assume the release of carbon is proportional to the depositional rates of biomarkers while considering the sea level was not constant and sea level was rising. Earlier time when the sampling locations were closer to the coast, the sampling sites should have higher depositional rates of carbon, and later when the sea level was higher the sampling sites were farther away from the coast. Therefore, even if the drainage basin was under the same condition, the depositional rates of carbon should be lower.

Lines 260-262: what about age model uncertainty?

Line 265: what process?

Lines 276-278: I would suspect the majority of permafrost derived carbon is reburied in sediments and this permafrost carbon is not the same source of carbon as discussed in the model. The permafrost you see in your sediment core is from the terrestrial land above water, while the yedoma carbon pool you talked about is below water and is stable.

Lines 278-279: This is the major factor affecting the estimates of permafrost carbon release back in time in many cases.

Lines 283-286: The marine side could not compensate the terrestrial carbon release. Marine takes CO₂ with minimal fractionation, therefore, if the CO₂ anomaly is due to terrestrial reasons, no matter what the marine side is doing, there should still be a significant shift in δ¹³C.

Line 301: the wetland formation in terms of CO₂ take-up is not discussed in this ms. If wetland could be such an important source of carbon in the sediment core in this study, then its impact on CO₂ level could not be ignored over geological time scales.

Line 302: It is rather a CO₂ sink than CH₄ source since it has net vegetation growth that takes up CO₂. But I do believe the abrupt CH₄ release could be due to wetland expansion globally.

Lines 307-309: But this does not explain the δ¹³C.

Lines 309-311: Although it is possible, but high accumulation rates of pre-aged biomarkers does not suggest shelf erosion and flooding as two dominant processes.

Lines 318-321: What about minimal changes in δ¹³C values through that time period?

Fig 1b: Apparently, back in time the Amur River drains into the ocean at a different location and how that would affect the consistency of your biomarker reconstruction and data interpretation is never discussed.

Fig 2h: What are age control points? How do you get the age, forams or plants?

Reviewer #3 (Remarks to the Author):

Winterfeld et al present a powerful dataset that attempts to constrain the export of permafrost carbon after the last glacial maximum. Understanding the fate of Arctic permafrost in a future warmer Arctic is critical for estimating atmospheric CO₂ concentrations and using paleoclimate records through deglaciations may be one way to constrain the fate of permafrost carbon. This manuscript presents high quality compound specific radiocarbon data of alkanolic acids, carbon that was captured during photosynthesis on land and aged in situ before being transported to the ocean. Additionally, chemical biomarkers that characterize processes occurring on land were also presented. To me, the truly unique aspect of this study is the coupling the biomarker analysis and radiocarbon data with paleoclimate modelling to estimate how changes in the eroded permafrost could impact atmospheric CO₂ concentrations and radiocarbon composition. While the paper presents a novel approach to close this important gap in our understanding of Arctic carbon, as currently presented, this paper falls short of convincing this reviewer that the export of thawed permafrost carbon is responsible for changes in the deglaciation atmospheric CO₂ record.

To me, this manuscript lacked a unified narrative. Based on the title of the manuscript, the paper focused on the mobilization of aged permafrost carbon during deglaciation. Based on the abstract, the paper focused how sea level rise during deglaciation influences the delivery of aged permafrost carbon during deglaciation. The results presented data on the changes in the composition of terrigenous material exported to marine sediments during deglaciation. The discussion started by saying this exported material is responsible for changes in the abundance and isotopic composition of atmospheric CO₂ during deglaciation. While it is entirely possible that the erosion of old permafrost carbon could be oxidized to change atmospheric CO₂, pulling the different parts of the argument together in a more seamless fashion is required. I believe that with mindful editing, this paper can unify these at times seemingly disparate visions for the dataset.

It was not clear what the sources of organic carbon were to the cores studied. In the introduction, the manuscript states that there is high riverine sediment discharge (line 68) that gets accumulated in the ocean. Yet, later in the paper it is speculated that sea level rise caused coastal erosion of old permafrost carbon that was subsequently incorporated into the sediment cores (lines 93-100). Is that simply a correlation or is it causation? Does this mean that riverine sediments are not important? Is there a way to estimate riverine based carbon from erosion based carbon? For example, would the composition of deep soil eroded from the coast have the same composition as permafrost carbon exported from the rivers? Are there mountains within the Amur River catchment area that could be responsible for the delivery of pre-aged carbon? At times during high discharge (Fig 2g), the age of the alkanolic acid also increases. Could this simply be an erosion signal from the mountains and not the mobilization of coastal permafrost that has thawed?

It is critical that we constrain how much permafrost carbon can affect the atmospheric CO₂ concentration. Since aged permafrost carbon was clearly mobilized during deglaciation and the concentration and isotopic composition of atmospheric CO₂ changed during that time, it is plausible that the permafrost carbon could have contributed to the recorded changes in CO₂. However, I have a few issues with this idea. The atmospheric concentration of CO₂ steadily increased during the Holocene, but it is also marked with periods of stability if not decline. For example, atmospheric [CO₂] rose during the HS1 yet stabilized during B/A. Similarly, atmospheric [CO₂] rose during the YD but stabilized during the PB. How do these results explain the changes in the atmospheric CO₂? The ¹⁴C composition of atmospheric CO₂ does not appear to go through abrupt changes that coincide with the export of terrigenous material, so I am not convinced that the rapid delivery of thawed and pre-aged permafrost carbon will be oxidized and significantly contribute to the atmospheric CO₂ record. For that to happen would

require many steps: export of pre-aged carbon (shown), full oxidation of said carbon (big assumption), all that oxidized CO₂ leaving the water column (assumed), thawed permafrost carbon in this region being representative of the rest of the world (assumed), and no other processes are leading to isotopic mixing of atmospheric ¹⁴C (assumed). While some permafrost carbon does get oxidized, as the authors state, the current estimates are between 2 and 88%. Mineral stabilization of organic matter protects a large portion of soil carbon from oxidation and this was overlooked.

Finally, mechanisms for the oxidation of aged permafrost were missing. Do the authors propose that the carbon undergoes microbial oxidation? Or would photochemical processing also change this material - albeit under limited light conditions due to high sediment load and latitude? The mechanism for this oxidation process could influence the ¹³C composition of the oxidized carbon. As the authors note, the atmospheric ¹³CO₂ recorded in ice cores illustrate changes in the ¹³C of CO₂ during the deglaciation and the ¹³CO₂ records do not reflect a release in terrestrial carbon. The paragraph acknowledging the lack of agreement with the ¹³C record (lines 280-289) then made me re-examine and question the agreement between the ¹⁴CO₂ record and the exported pre-aged permafrost carbon. But it also highlighted an additional uncertainty for me: what is the ¹³CO₂ composition of photo-oxidized organic matter? CO₂ evolved from systems that are not carbon constrained undergo biological fractionation of ¹³C, which leads to a depleted ¹³CO₂.

By my reading of Figure 3, the ¹⁴C values of the alkanolic acids in the two cores did not completely tell the same story. The LV28-4-4 samples were more pre-aged at times that didn't coincide with thawing the same as the SO178-13-6 samples (Figure 3). What could explain this difference in the amount of pre-aged material? Was the biomarker composition (or bulk composition - C/N, ¹³C_{org}, etc) similar between the two cores? Could this disparity between the two cores be related to the sample size of the ¹⁴C samples? I noticed that the LV28-4-4 samples were smaller and could therefore contained more exogenous carbon?

Other notes:

Figure 1: I found these maps confusing. The current shoreline was not evident and due to the greyscale contrast of the seafloor, the nearby subduction was a focal point. Please edit these maps to clearly highlight the current sea shore and the study area.

Table 1 + Supplementary: I appreciated the efforts in generating accurate and precise radiocarbon measurements. This does not always happen. Since the R₂ was 'ignored' for the C₃₀ (¹⁴C free material) how was the uncertainty of the modern contamination estimated? Does an uncertainty of blank of 0.08 ug C seem reasonable based on other studies that have assessed ¹⁴C contamination in CSRA samples? This seems small to me - as does the uncertainty of the CSRA samples given in Table 1.

Author responses to reviewer comments of the manuscript entitled “Deglacial mobilization of pre-aged terrestrial carbon from thawing permafrost” (NCOMMS-17-27886) by M. Winterfeld and co-authors

We thank all three reviewers for their comprehensive reviews and we greatly appreciate the constructive and helpful comments. All comments were carefully considered and addressed, and most of the suggestions were incorporated in the revised version of the manuscript. We feel that this new version of the manuscript was improved, particularly the modeling part.

In response to the reviewers’ comments, we made a number of changes, the most important of which are:

- All reviewers had critical comments regarding our model assumptions. We consider the modelling exercise, however, an integral and novel part of the manuscript (as also stated by reviewer 3: *“To me, the truly unique aspect of this study is the coupling (of) the biomarker analysis and radiocarbon data with paleoclimate modelling to estimate how changes in the eroded permafrost could impact atmospheric CO₂ concentrations and radiocarbon composition.”*). We therefore improved the description of our model and its assumptions, and clarified our conclusions drawn from the model output to avoid further misunderstandings. In response to the criticism raised, we have slightly modified the assumptions – in detail the upscaling – used for our carbon cycle simulations. Changes mainly affect the estimates for the amount of carbon lost from Yedoma during the deglaciation (259 Pg (Strauss et al. 2017) instead of 285 PgC in the original manuscript), and for the assumed remineralization rate of the mobilized carbon (66% (Vonk et al., 2012) instead of complete remineralization (100%) in the original manuscript). These changes combined with an estimate of the flooded shelf area resulted in a release of 85 Pg of carbon from shelf-flooded permafrost during Termination I, which is a much lower estimate than in the initial draft (in which we have stated to estimate a maximum (or upper) bound of 170 PgC) and a smaller contribution to the total change in atmospheric CO₂ concentration and its $\Delta^{14}\text{C}$, but did not affect the overall conclusions. The use of these different values also required a revision of Figures 4 and 5.
- We re-organized the chapter describing the modeling exercise in order to explain the approach and the assumptions inherent in it.
- We improved the description of our coring location and explained in more detail why these sediments are representative of both, river discharge and coastal erosion due to sea-level rise
- We added a figure to the supplement information displaying biomarker concentrations to illustrate that peaks in accumulation rate are also peaks in concentrations (new Supplementary Figure 1).

Below, we repeat the reviewer comments and reply to the concerns one by one with our responses in **blue**. Some concerns like the process of erosion of permafrost coasts during deglacial sea-level rise were raised by all three reviewers. We therefore replied in detail the first time the issue came up and referred to this explanation when replying to later comments.

Reply to Reviewer #1:

1. Reply to general comments

Overall, as reviewer #1 correctly states, the main focus of this manuscript is to provide physical evidence for the mobilization of strongly pre-aged carbon derived from permafrost deposits during the last deglaciation. The second part of the manuscript presenting the simulation results of a carbon cycle model based on the timing of increased accumulation of terrigenous organic matter and its ^{14}C ages as found in this study provides a first data based estimate of a possible contribution of permafrost-derived carbon to the three distinct increases in atmospheric CO_2 during the last deglacial, by estimating maximum values and serving to explore the order of magnitude the mobilization of permafrost carbon would have had on atmospheric CO_2 and $\Delta^{14}\text{C}$. So far, modelling studies exploring the possibility of deglacial CO_2 contributions from warming permafrost did not rely on physical data, but assumptions (e.g. Crichton et al., 2016; Köhler et al., 2014). The aim of our model approach is to give the reader further context of possible implications of our findings with regard to the deglacial carbon cycle. To refine our understanding, further and more detailed modelling work will be needed as well as more compound-specific ^{14}C data from locations bordering the Siberian shelf seas to constrain CO_2 contributions from degrading permafrost, however, this cannot be achieved within the scope of this manuscript.

We are confident that our model approach is a valid first estimate for the contribution of permafrost-derived carbon to these deglacial CO_2 peaks and that it is an integral part of our study providing context to the reader. However, the reviewer's comments showed us that this was not always clear and easy to understand in the manuscript.

1.1 *"The modelling component of the study tries to establish physical erosion of Yedoma deposits as the principle mechanism of permafrost carbon mobilization during the melt-water pulses. The much of the modern Yedoma deposits do boarder the Eastern arctic sea (Strauss et al. 2013), and up to 80% of the flooded shelf is estimated to have been part of the Yedoma region during the LGM (Walter et al. 2007). However, the authors appear to have overestimated the amount of carbon likely held in these flooded Yedoma deposits, and do not establish how that coastal erosion could have mobilized such a large fraction of this organic matter. The modern Yedoma deposits are estimated to contain 181 PgC (Hugelius et al. 2014). If 80% of the 1.9 million km² of Eastern arctic shelf was Yedoma region in the last LGM and if the region had the same carbon density of the modern deposits, then the area would contain ~200PgC. This estimate is smaller than the authors estimate (285 PgC) but the more important issue is how much of the carbon would be mobilized when the region flooded?"*

Reply 1.1: Calculating the size of the carbon stocks possibly stored in Yedoma-like deposits on the now-flooded Siberian shelves strongly depends on the understanding of today's Yedoma deposits and its carbon density. We therefore do not agree that our estimate of 285 PgC is overestimated per se as we used published values for carbon content, bulk density, and average deposit height from Strauss et al. (2013) to calculate it. However, there are different approaches of how to calculate the carbon stocks of this area. It should also be noted that today's estimates are associated with large uncertainties in the range >100 PgC, and we stress that the estimate of ~200 PgC given by the reviewer and our estimate of 285 PgC are within this range of uncertainty.

In order to address the criticism raised, we used carbon stocks from a recently published synthesis on Yedoma deposits by Strauss et al. (2017) for a revised calculation

of the carbon stocks present in Yedoma-like deposits on the flooded shelves. In this recent compilation, the carbon stock of Pleistocene Yedoma deposits (including the now flooded Arctic shelves) was given as 657 ± 97 PgC. If we subtract today's carbon stocks of the Yedoma domain of 398 PgC, including thermokarst and tabular deposits, which have been Yedoma deposits before warming and thawing since the last glacial, we obtain an estimate of 259 PgC likely stored in now flooded Yedoma deposits. This number is between our original estimate of 285 PgC and the estimate from reviewer #1 of ~ 200 PgC. Considering the associated uncertainties with these kind of estimates, we used the new estimate of 259 PgC. We then accounted for the fraction of the shelf area flooded between 18 and 11 kyrs BP from -130 m to -50 m of present day sea-level (50% of the shelf) based on the global sea-level curve from Lambeck et al. 2014 resulting in 129.5PgC (50% of 259 PgC) stored in the flooded area. Originally, we then used a remineralization rate of 100% in order to calculate the upper limit of possible CO₂ contributions. This was probably not well explained in the manuscript and therefore raised some questions by the reviewers. In the revised version of the manuscript we now include a carbon remineralization rate after erosion of $66 \pm 16\%$ taken from Vonk et al. 2012 resulting in 85 PgC being mobilized between 18 and 11 kyrs BP. It should be noted that these assumptions, as well as the model-inherent uncertainties pertaining, e.g., to the airborne fraction of CO₂ released to the atmosphere (8% at the 8 kyr-timescale (Supplementary Figure 3a) in our model is at the lower range (6-16%) of values commonly obtained by carbon cycle models), are associated with large uncertainties. Therefore the resulting estimates of how much permafrost-derived carbon could have contributed to the three phases of rapid CO₂ increase at 16.5, 14.6, and 11.5 kyrs BP provides an estimate of the order of magnitude of this effect. Exact numbers should be treated with caution.

The edits of the manuscript regarding these calculations affected lines 200-250.

1.2 *"... The authors assume that all of the carbon in the flooded region was mobilized, however they do not provide evidence to support this assumption. Certainly some of the carbon would be mobilized through erosion of sea-cliffs but how deeply where cliffs eroded? Yedoma deposits are characteristically 10s of meters thick assuming that the coast was eroded by that depth of soil over such a large area seems extraordinary. Presumably much of the coast would have been relatively passively flooded resulting in the wide-spread subsea permafrost seen in the region today (Shakhova et al. 2010). That is, much of the LGM permafrost carbon may still be frozen in the permafrost of the East Siberian Arctic shelf, below sea-level but **kept frozen** by ocean temperatures below 0°C.*

Reply 1.2: First we would like to note that the existence of sub-sea permafrost only refers to the fact that the deposits remain frozen throughout the year but does not mean that they are carbon-rich deposits. Yedoma deposits are typically developed on top of frozen sandy sediments, which are much less susceptible to thawing and mobilization and, unlike the overlying ice- and carbon-rich Yedoma, have the potential to remain in place during sea-level rise.

The process of thermo-erosion of today's permafrost coasts was studied intensively over the last decades (e.g. Dupeyrat et al., 2011; Lantuit et al., 2011, 2012; Günther et al. 2013; Barnhart et al., 2014; Hoque & Pollard 2016). There is no evidence that this process was different during the deglacial warming and sea-level rise from what we observe today. The idea of non-erosive flooding and preservation of ice-rich permafrost in sub-zero ocean water (as stated by reviewer #1 citing Shakhova et al., 2010) is in

principle possible, but has not been observed anywhere. Where ice-rich permafrost is eroded, the entire ice- and carbon-rich stratigraphy is eroded (the upper 10s of m) and mobilized. Even where ice-rich permafrost extends below the seabed (e.g. central Laptev Sea), the ice-rich permafrost is degraded within a few centuries after inundation, and the sediments are mobilized by wave turbation and ice gouging (Shakhova et al. 2017). There is no evidence for high organic carbon contents preserved in sediments below the eroded strata (e.g. Winterfeld et al. 2011). In contrast, several recent studies described the role of deglacial sea-level rise flooding the East Siberian Shelf in eroding and distributing the permafrost deposits that covered the flooded areas (e.g., Bauch et al., 2001; Keskitalo et al., 2017)

Periods of high accumulation of terrigenous material not only coincide with periods of increased sea-level rise, they are a result of it. Based on the global sea-level curve, the average sea-level rise could have been 25-35 m/kyr during the B/A and 10-20 m/kyr during the PB warm periods. These rates are comparable to the highest local rates of sea-level rise observed today, which result in drastic modern erosion rates of up to 20 m of coastline along Yedoma coasts. Therefore, we consider the assumption of a complete erosion of these ice- and carbon-rich deposits reasonable. We acknowledge that this might not have been clear to the reader and we edited the respective paragraph (line 148-149)

1.3 *“I agree that the LGM permafrost carbon pool could have played a key role in the spikes in atmospheric CO₂ seen during the melt-water pulses. However taking into account the response of the inland permafrost carbon pool to the abrupt warming during these intervals also seems crucial.”*

Reply 1.3: We agree that responses from inland permafrost degradation during these warm phases might have played an important role as well. However, when analysing the terrigenous organic matter buried in marine sediments we study the part of the organic matter that escaped remineralization and was transported to and deposited in the ocean. Information about the respired organic matter on land is not recorded in marine sediment cores.

The processes of inland thaw are much more difficult to reconstruct from our sediment records, as we rely on evidence for changes in landscape development. Thawing inland permafrost is expected to first result in a wetting of the landscape, i.e., developments of lakes and extensive wetlands. We use biomarker evidence for a change in wetland extent in the Amur river basin (see lines 185-191) and find that this change occurred gradually with a maximum occurring later during the deglaciation and towards the early Holocene. During this time period, we also did identify pre-aged organic matter likely exported by the Amur River during times of high river discharge, which we interpret to represent inland permafrost degradation and erosion from inland permafrost soils. However, our sediment-based evidence clearly shows the more rapid and drastic occurrence of permafrost carbon mobilization during the phases of rapid pulses in sea-level rise, which was initially unexpected but suggests, together with the evidence for changes in landscape cover, a stronger impact on the atmospheric CO₂ records by sea-level than by inland thaw.

We do not neglect the potential of inland thaw processes to release CO₂ (and CH₄). However, including the terrestrial response of permafrost degradation during the deglacial warming would require a more sophisticated modelling approach than we can do in this study, in particular regarding the change in carbon stock in the inland areas

during the transition from permafrost to non-permafrost. This information is currently not available.

Specific comments:

Units: Both kyr and ka are used. I prefer ka but you may want to consult Nature's style guidelines.

Both, ka and kyr are published in Journals of the Nature Group, and we changed the units to kyrs.

Line 62: Rework this sentence to better orient the reader. Most will not know that the Amur river forms the much of the border between China and Russia.

We would like to use caution with using country names in our geographic description, as we consider the geographic location the only relevant information in context of our study of past climate conditions.

→ changed sentence beginning in line 71

"The Amur River basin is the largest catchment in East Asia. The region..."

Line 91: I recommend writing out Pre-boreal everywhere instead of using an abbreviation.

→ changed to Pre-Boreal throughout the manuscript

Line 98: Change "rather" to "relatively"

→ done

Line 105 to 106: Weaken this statement, i.e. 'must have' to 'likely have'

→ done

Line 198: Change "rather" to "relatively"

→ done

Line 196: Change "Arctic region" to "permafrost region". Most of the LGM permafrost was well outside the Arctic. The Laurentide ice sheet extended below 40°N, and the permafrost region extended well south of the ice sheet edge.

→ done

Line 221: Write out "Ref. 3"

→ done

Line 278 to 279: This sentence does not make sense. Total loss and rate of loss are being conflated.

→ sentence was rewritten (line 301-305)

"Furthermore, some of the permafrost-derived carbon is found in our sediment cores in the Okhotsk Sea, underlining the uncertainties associated with the estimated remineralization rate of 66%, which is potentially too high. The current range of estimates for carbon loss upon thaw varies from 2 to ~80% of the initial carbon stock^{35,36}."

Line 308: Change "the last couple of years" to "recent years"

→ done

Line 494: Change "While in" to "In"

→ done

Line 497: Change comma to period. Re-write next sentence to "While in our additional simulations it was prescribed"

→ done

Reply to Reviewer #2:

The main concerns raised by reviewer #2 relate to the lack of novelty of our study compared to a previous publication by Tesi et al., 2016, the choice of sampling locations, which he/she doubts being representative of Amur river discharge, and the assumptions made in the modelling exercise. The latter comments were already mostly addressed above in response to reviewer #1. Below, we provide our replies to the concerns of “novelty” and “sampling sites”.

2.1: *“[...] However, I also find this manuscript has many issues. For example, many discussion and conclusions are not well supported by either the data or the way of data interpretation using proxies. I find it hard to separate the novelty and validity of this study and how the conclusions could contribute to our current knowledge on the last glacial. Tesi et al., 2016 (Nature Comm) presented a case study suggesting massive remobilization of permafrost carbon during post-glacial warming. Although there are some issues involved in that manuscript, based on my opinion, that manuscript has answered most of the major conclusions that have been drawn in this study. [...]”*

Reply 2.1:

First of all, as reviewer #2 feels that our discussion and conclusion do not support our interpretation, there seem to have been a lack of clarity in some of our statements. We reviewed the discussion and conclusion sections to rephrase and/or add sentences in order to make our line of thinking better to follow.

Secondly, we do not agree with reviewer#2’s opinion regarding the novelty of our study. It is indeed correct that Tesi et al., (2016; Nat. Comms.) provided evidence of terrestrial carbon mobilization at the beginning of the Holocene warm phase, but the presented data only cover the time period after 11.7 ka BP and hence do not cover the glacial-interglacial transition during which major changes in carbon transfer between land, ocean, and atmosphere are expected. Our study, moreover, is the first to provide data on the age of this mobilized terrestrial carbon and the timescales of carbon release from degrading permafrost during the entire deglaciation, which is the most recent, drastic climate warming phase.

We provide direct evidence of massive deglacial (*before 11.7 ka BP*) mobilization of strongly pre-aged terrigenous carbon by compound-specific radiocarbon dating of land-derived biomarkers deposited in rapidly accumulating sediments of continental margins adjacent to the Amur River basin, one hotspot of deglacial permafrost thaw. Tesi et al., in contrast, only showed the presence of old carbon, but, since they worked with bulk ¹⁴C ages alone, could not provide unambiguous data on the age of the terrestrial portion of it.

To our best knowledge, we are the first to provide unambiguous evidence for pre-aged organic matter remobilization from a region that lost most of its permafrost coverage during the deglaciation (again, in contrast to Tesi et al., who worked off the Lena river, whose drainage basin remains almost entirely permafrost covered until today; changes in this drainage basin thus must have remained rather less dramatic than in the Amur basin). We link high accumulation rates of terrestrial biomarkers to the melt water pulses 1A and 1B, and biomarker ages during these events ranged from 5,000 to 10,000 years at the time of deposition. In summary, the Tesi et al., 2016 study does not cover

the last glacial-interglacial transition, does not provide ages for terrestrial organic matter based on analysis of terrigenous biomarkers (only modelling based on carbon isotopes), and the study location is situated in an area where continuous permafrost is prevailing until today. Therefore, our study certainly adds new knowledge to our understanding of carbon mobilization during the last deglacial. In addition, our carbon cycle model provides data-based estimates of the order of magnitude of the impact that these processes could have had on the deglacial increase of atmospheric CO₂ from pre-aged sources.

Major issues:

2.2: *“1) The sampling locations and how the sedimentary records are linked with permafrost thawing are susceptible. Based on figure 1, the sampling locations are on the east of the Sakhalin Island and are pretty far away from the current mouth of the Amur River. Currently, the Amur River drains into the north side of the Strait of Tartary west of the Sakhalin Island. To what I know, it may have minimal effect to where the samples were taken. However, during last glacial period, the Strait of Tartary was above water and the river mouth was inland. Therefore, the river mouth should be at a different location. The Amur River most likely drains to a location north of the current one.*

Based on figure 1, the sampling location, especially SO178-13-6 was close to the coast and had a shallow water depth back in last glacial period. However, with sea level rise, the water depth went deeper. It is also worth note that the water depths of these two cores are different, with core LV28-4 much deeper. Therefore, it is very likely that core SO178-13-6 was largely affected by coastal processes since it is closer to the coast. When the water depth went deeper, it definitely has less effect from coastal erosion. However, since these two sampling locations are pretty far away from the river mouth, and may not have direct linkage with the river discharge, I highly doubt the sedimentary record is representative of what happened in the drainage basin. Instead, it may represent what happened on the Sakhalin Island since it is closer. Furthermore, I don't understand why authors picked two cores and only did minimal work on the second core. The authors could have done more CSRA on one core to improve the resolution of radiocarbon analysis.

When I read through the manuscript, I see one major focal of the manuscript is to distinguish carbon sources of coastal erosion and hinterland inputs through river transport. However, based on my concerns, I highly doubt the validity of the discussion.”

Reply 2.2:

We described in the introductory part of the manuscript (lines 68-72) and the methods section (starting line 351) how the circulation in the Okhotsk Sea works and how the material delivered by the Amur River to shelf north of the Sakhalin Island is rapidly and efficiently incorporated into the southward-flowing East Sakhalin Current (ESC), which transports surface and deep waters from the northern shelves to the Kuril Basin (Ohshima et al., 2004). The topic of entrainment and activation modes of lithogenic, and biogenic particles, and POC and DOC into the Dense Shelf Water and their subsequent transport into mesopelagic Okhotsk Sea Water has been subject to extensive research efforts in the past ca. 20 years and a number of studies have ascertained the assumptions we used in our paper, based both on modeling and instrumental data. We refer, amongst other studies to e.g., the journal special issues in Journal of Geophysical Research – Oceans (vol. 109, C9; cf. Ohshima and Martin, 2004), or more recently in

Progress in Oceanography with the topic “Biogeochemical and Physical Processes in the Sea of Okhotsk and the Linkage to the Pacific Ocean.” (cf. Nishioka et al., 2014). However, we understand we may not have referenced these works sufficiently enough in our initial manuscript version, thus have rectified this deficiency in the revised version. We also take the liberty to refer to a recent paper by some us (Lembke-Jene et al., 2017) that linked sedimentological hinterland sources with downstream transport mechanisms along the Sakhalin margin into the North Pacific for the last deglacial transition, which is the time frame of this contribution.

In short, during the sea-ice season in fall and winter, Dense Shelf Water (DSW) is formed and flows south along the Sakhalin margin, transporting high concentrations of organic matter, lithogenic particles and suspended matter that are entrained by vigorous tidal mixing on the northwestern shallow continental shelf into a highly turbid water layer (Nakatsuka et al., 2004a, b). Combined with discharge from the Amur River these materials rapidly accumulate along the East Sakhalin margin (Nakatsuka et al., 2004a, b). This particular circulation system in the Okhotsk Sea favors a transport of suspended and particulate material along the Sakhalin margin with the East Sakhalin Current and leads to maxima in deposition of riverine material at the chosen sites (Seki et al., 2006, 2014) and riverine material is deposited in the Okhotsk Sea sediments (e.g. Seki et al., 2014, Ternois et al. 2001, Seki et al., 2012, Gorbarenko et al., 2010, Gorbarenko et al., 2014, Lembke-Jene et al. 2017).

Secondly, the reviewer entertained the opinion that sites closer to the Amur River mouth, e.g., in the Tartar Strait, would more adequately represent Amur outflow. In this regard, we refer to the extensive pre-site survey we have admittedly not detailed sufficiently in our first version. We have rectified this here for the reviewers’ and editor’s information by providing additional site and regional information here in our reply, which were published previously in the publicly available cruise reports (GEOMAR Report Series). Briefly, during cruise LV28 and subsequent cruises (LV29, GE99, LV31, etc.) we carried out seismic works along the Sakhalin continental margin, (including extensive IODP pre-site survey lines), high-resolution sub-bottom profiling and coring activities closer to the mouth of the Amur river, including proximal basins like the Derugin Basin, Kashevarov Bank, N’ Sakhalin, including a multi-day survey directly in the Tartar Strait (cf. Biebow et al, 2003). We also carried out surface sediment and core sampling in all of these areas.

From the cruise narrative during LV29 (Biebow et al., 2003):

“After having finished the sediment core transects, we went around the northern tip of Sakhalin into the Sakhalin Gulf (Amur River estuary) on July 9th (2002) in order to take water and sediment samples there. Amur River is the largest source for fresh water and sediment of the Okhotsk Sea and the 4th largest Siberian river. Apart from that, Amur River is the only of the large Siberian rivers which [sic] does not flow into the Arctic Ocean. We were mainly interested in the effect the Amur waters have on sea-ice formation and productivity of the Okhotsk Sea. We mapped the area of the Amur River estuary two days and carried out extensive water sampling. The fact that Amur River transports large amounts of sediment into the Okhotsk Sea is visible even from the vessel. Unfortunately, it quickly became clear that exclusively sand up to coarse gravel is deposited in the Sakhalin Gulf. Our attempts to directly sample the Amur sediments at three stations thus were unsuccessful as the cores could not penetrate the sand layers. We recovered only fist-sized pebbles. Fortunately, the coring equipment was not damaged.”

In essence, the few sites we were able to find directly in Tartar Strait / the Sakhalin Gulf did not yield sediments conducive to coring operations due to high current energy from riverine outflow and tidal currents. Repeated coring attempts in these locations only provided coarse sands to gravel-size, likely relic sediments and dropstones. Also in our sub-bottom profiling works we found a number of presumed small-scale paleo-river discharge channel fills that most likely stem from last glacial time intervals. Slightly more distal locations do receive finer sediment material (needed for biomarker analyses) but are hampered by strong lateral transport with the East Sakhalin Current further south (to the site locations we chose). In fact, deglacial to Holocene sediment sequences in these more proximal areas do not exceed 5 cm/ka, an order of magnitude lower than the ones we used for our reconstructions. These sites would also have not yielded enough sediment for the sampling strategy we used for our study.

Third, it is incorrect that the two cores were taken at “much different” water depth. In fact, core LV28-4-4 was taken at 674 m water depth (shallower), while core SO178-13-6 stems from 713 m water depth. Both of these depths are located on the continental slope, and changes in sea-level from the glacial to the Holocene would not move these locations towards a depth with a different sedimentary regime. As detailed above, both core locations are primary depositional sites for fine-grained terrestrial material supplied by the Amur river and by suspension processes at shallower water depths. The map provided as Figure 1 of our manuscripts displays in panel b also the inferred coastline during the Last Glacial Maximum, nicely illustrating that sea-level change did not lead to a substantial change in the geographic and bathymetric context of the core locations. If the relative location to the shore were the controlling factor on the accumulation rate or terrestrial organic matter, we would expect to see maxima during sea-level low-stands with gradual decreases during sea-level rise. Our observations are very different, i.e., low accumulation rates of terrestrial organic matter during the sea-level lowstand and pronounced maxima during the phases of rapid sea-level rise.

Furthermore, we chose to work on these two sediment cores because of their different accumulation rates in the Holocene and deglaciation. Core LV28-4-4 has a relatively lower resolution during the deglacial than core SO178-13-6. Despite this difference in accumulation rates and resulting temporal resolution of the records, both cores show similarly pre-aged terrigenous biomarkers despite their differences in resolution. This agreement is further evidence for the validity of our interpretation of these records as reflecting processes in the Amur river basin and the Okhotsk Sea shelf/coast. It would certainly be good if additional compound-specific radiocarbon ages were available for both cores; however, these cores had previously been studied for a number of parameters, and due to the large amount of sediment material needed for compound-specific ^{14}C dating we were limited to only a few available intervals.

Changes made in response to this comment can be found in the Methods section (line 369; 380-395).

2.3: *“Based on my knowledge, I don’t think the modeling part makes sense at all. The model was based on the assumption that ~60% of the permafrost deposited on the Arctic shelf could have been released during last glacial time because the sea level raised by 60% from ~ -120m to ~ -50m. However, I don’t understand the logic here since whenever sea level rises, especially when rise rapidly, this ~60% of the permafrost deposition was supposed to be submerged under water and be buried quickly by later deposition from terrestrial sources. I got further confused when authors assign this assumed ~170PgC release into three time periods based on peak periods of biomarker accumulation. I was*

totally lost when authors have the whole modeling discussion based on such assumptions. In addition, in terms of modelling and real data, it should always been modeling based on real data, other than data interpretation based on modeling. In this case, the number of ~170PgC release was calculated based on a simple model and this number has been widely discussed and applied through the majority of the discussion. Therefore, I don't think the discussion is valid at all."

Reply 2.3:

We would like to refer to our reply 1.1 (above) regarding the carbon cycle model and reply 1.2 (above) regarding permafrost coast erosion and sea level rise. As mentioned above, we re-phrased parts of the model description and model interpretation to clarify it (see lines 200-250 and 282-290).

The reviewer seems to have misunderstood our approach of how we employed the model in our study. We used the results from our sediment cores analyses to assign an age to the remobilized carbon, and consequently to possibly released CO₂, as well as to determine the timing of the release based on the increased accumulation rates that we find the sedimentary record. Prior to our study, no such data have been available, which is why all previous studies attempting to quantify the effect of permafrost thaw on atmospheric CO₂ and $\Delta^{14}\text{C}$ have relied on assumptions regarding timing and isotopic composition of the carbon release (e.g. Köhler et al., 2014, *Nature Communications*; Crichton et al., 2016, *Nature Geosciences*).

We observe three peaks in the accumulation rate providing evidence for release of permafrost carbon. They coincide with the times of rapid increase in atmospheric CO₂ as described by Marcott et al., 2014, and we used these peaks to fix the time intervals of enhanced remobilization of permafrost-derived carbon and subsequent cycling of that carbon in our carbon cycle model. Their relative magnitudes furthermore served to scale the amount of the carbon release in our model, and measured radiocarbon values of terrigenous biomarkers were used to derive the impact on $\Delta^{14}\text{C}$ of CO₂ in the atmosphere.

While our study provides a solid data base for the timing and isotopic composition, we are left with the necessity of still making assumptions, albeit based on data-based estimates from the literature, regarding the carbon stock on the now submerged shelves. However, as mentioned in reply 1.1 we revised the numbers for the carbon stock that could have been remobilized during the deglacial and other parameters of the carbon cycle model to account for the uncertainties, and we re-phrased parts of the discussion (lines 289 and following, and 321 and following). We emphasize again that we fully acknowledge the uncertainties associated with these estimates. Nonetheless, the carbon cycle model is an essential part of our manuscript, because it puts our findings in context in regard to the magnitude to which permafrost carbon could have affected the carbon cycle during the deglacial transition.

Minor comments:

Line 54: Does not follow the logic of the previous sentence and is a little bit odd here.

→ We rephrased the preceding sentence so that the section now reads (lines 55-60):

“Until now, assessments of the susceptibility of permafrost to degradation and assessments of future and past feedbacks rely mainly on simulation scenarios, including assumptions based only on indirect estimates of the age of permafrost-derived old

carbon, but physical data are so far lacking^{3,6}. Thus, the age, timing, and quantity of terrigenous carbon remobilized during the last deglaciation is largely unknown.”

Line 55: The general audience may not know the exported material includes particulate and dissolved. Please make it clear.

→ added “In contrast to dissolved organic matter, ...” at the beginning of this sentences (Line 61)

Lines 56-59: This is not clear on how radiocarbon ages of terrestrial carbon can help answer carbon-climate feedback.

→ Because only fossil, i.e. very old permafrost-derived carbon would affect the carbon-climate feedback. That means the mobilization of old permafrost carbon and its burial in ocean sediments are prove of old carbon being available for remineralization and thus likely affecting carbon-climate feedback.

Rephrased to read: “In contrast to dissolved organic matter, particulate organic matter is the only fraction of exported material that can be preserved in sediments and records past climate and environmental conditions. Accumulation rates and radiocarbon ages of terrigenous carbon in sediments adjacent to areas of permafrost degradation thus allow a data-based evaluation of the permafrost carbon remobilization and associated carbon-climate feedback, including its contribution to deglacial CO₂ rise.” (lines 61-66)

Line 61: Don't think it is possible considering the old ages and mixing of variable carbon ages during export.

→ Reconstructing the vegetation and climate history on land based on biomarker analyses and compound-specific isotope analyses (e.g. $\delta^{13}\text{C}$, $\Delta^{14}\text{C}$, $\delta^{18}\text{O}$, $\delta^2\text{H}$) of terrigenous organic matter buried in marine sediments offshore of large river mouth is well established and has proven to be a valuable approach despite possible mixing of variable carbon ages during transport (e.g. Schefuß et al. 2016, *Nature Geoscience*; Gustafsson et al., 2011, Seki et al., 2003, Håggi et al., 2016). The commonly accepted view is that the terrigenous organic matter accumulating in marine sediments is composed of a mix of organic matter derived from pre-aged and modern sources. While the pre-aged proportion defines the average age of the material deposited, a rapidly cycling contribution will carry a climate and vegetation signal, which is damped by the pre-aged contribution (see, e.g., Galy et al., 2015, *Nature*).

Line 64: change Amur to Amur River

→ done

Lines 68-70: In this case, the sediment core, even terrestrial biomarker radiocarbon age doesn't necessarily represent terrestrial material discharged from the Amur River.

→ We do not say the organic matter is only derived from the Amur River. It is a mixture of sources and at different time periods during the last 17 kyrs the Amur River might have been a more dominant source. Please also refer to reply 2.2 above.

We slightly modified the sentence to make clear that the material transported from the shelf areas also entrains terrigenous matter. (line 76)

Line 70: As I stated in the major comments, the sampling cores are one of my biggest concern here as back in time when sea level was lower the Strait of Tartary was above sea level and Amur River drains across Sakhalin and may largely affect the sampling locations in this study. However, during deglacial time and the Holocene when the sea level was higher with the formation of the Strait of Tartary, the Amur river drains into the Strait of Tartary directly and doesn't affect the sampling locations at all. For

example, if you check on the current world river map, the Amur River drains into the north part of the Strait of Tartary. This may also explain the high sedimentation rates and OCAR at some time periods. Therefore, the OCAR may not represent how climate shifts affect carbon exports, but more representative of drainage basin size that affects these sampling locations.

→ Please refer to reply 2.2 above for a more detailed response. It was shown by several other studies that Okhotsk Sea sediment core can be used to reconstruct the paleoclimate of the Amur River catchment since the last glacial (e.g. Seki et al., 2012, Gorbarenko et al., 2014). References to these publications have been provided in the manuscript, and we invite the reviewer to consult these publications for a discussion of the sedimentary regime of the locations where our cores were recovered.

Lines 70-71: These two sediment cores are far away from each other and apparently receive different materials with sorting of material during transport. There has been study that hydrodynamic sorting has an effect on the age of carbon.

→ This is a very general remark and it is difficult to know what exactly the reviewer is referring to without an example or a reference. There are studies which could show that hydrodynamic sorting has an effect on the composition and quality of organic matter (e.g. Bergamaschi et al., 1997, Vonk et al., 2010, Tesi et al., 2016). Usually finer terrigenous organic material would be more degraded and therefore associated with older soil-derived organic matter while coarser organic material appears to be fresher and thus younger associated with plant debris. Hydrodynamic sorting has an effect along steep gradients, e.g., from very shallow coastal waters towards deep offshore regions, where the transported material (including the organic matter) decreases in grain size. However, in this particular study the sediment cores are in 674m (LV28-4-4) and 713m (SO178-13-6) water depth and due to this pelagic zone the material accumulated here is already in the fine fraction and no differences in ^{14}C due to hydrodynamic sorting are expected here. Differences in age between biomarkers at two core sites would be expected only if they strongly differed in sedimentary regime, including oxygen levels or sediment input (e.g., Mollenhauer et al., 2003, 2007; Mollenhauer and Eglinton, 2007; Kusch et al., 2010). In our case, despite the slightly differing water depths, both cores are at present under the influence of the same water mass, highly turbid Dense Shelf Water, as evidenced by a number of recent studies (e.g., Shcherbina et al, 2004). Moreover, compound-specific biomarker ages between the two cores show generally the same pattern with strongly pre-aged terrigenous biomarkers during the deglaciation and younger biomarkers near the surface. Our study does not interpret the absolute age differences between the cores. Instead, we focus on the generally consistent pattern.

Line 73: Could also be from the erosion of petrogenic sources?

→ Long-chain *n*-alkanes, which are derived from terrestrial plants, could potentially be contaminated by *n*-alkanes derived from petrogenic sources (Pearson and Eglinton, 2000) and would thus bias the compound-specific radiocarbon analysis performed to identify permafrost-derived organic matter. Unlike long-chain *n*-alkanes, sedimentary long-chain *n*-alkanoic acids are expected to be free of contributions from petrogenic sources (Eglinton et al., 2002) and therefore be more representative of average terrestrial residence times of vascular plant biomarkers (Kusch et al, 2010).

We added an explanation of the use of *n*-alkanoic acids compared to *n*-alkanes and branched GDGTs in the methods section (Line 419-426).

Line 74: BrGDGT is only representative of soil erosion, not fresh material inputs and is highly biased by soil development.

→ We would like to point out that our study does not address fresh organic matter input but rather erosion of thermally degraded permafrost deposits/soils.

When permafrost soils thaw and become destabilized, particulate organic matter is remobilized, transported and buried in sediments. We use brGDGTs here to investigate the remobilization of permafrost-derived soil organic matter. The fact that their concentration and accumulation rate records are very comparable to those of n-alkanes as biomarkers of higher land plants suggests that indeed our biomarker records are reliable indicators of input of terrestrial organic matter stored in permafrost deposits, which contain the organic remains of plants and soil microbes.

No action taken.

Line 76: what's the purpose of analyzing different biomarkers for concentrations and radiocarbon age? Could have been much more helpful if fatty acid concentrations are reported as well.

As pointed out in the revised methods section, the quantification of n-alkanes is analytical more robust, as an apolar internal standard as used for quantification of n-alkanes, after extraction and separation of apolar compound class is less prone to be partially retained on chromatographic columns. Using n-alkane concentrations and accumulation rates as records of terrigenous organic matter input is a well-established method in paleoclimate studies (e.g., Schefuss et al., 2011, Nature, 2016, Nature Geoscience).

In contrast, n-alkanoic acids need to be methylated and can only be analyzed as fatty acid methyl esters on GC-FID, which introduces uncertainty during the wet chemical preparation. As a result, their quantification down core is more susceptible to analytical variability, which often results in noisier records.

We also point out that both two biomarker records (alkanes, brGDGTs) show similar trend, which is taken as an indication of robustness. Moreover, we refer to the literature showing that typically, concentration records of n-alkanes and n-alkanoic acids co-vary (e.g., Ohkouchi et al., 1997). See changes in methods section, lines 419-426.

Lines 87-88: How is this compared with the MAR and OCAR. I suspect they all have the same trend and all the trends are ultimately controlled by the sedimentation rates, which is linked with material inputs. Further, material inputs are mainly affected by the offshore distance and riverine inputs.

In the revised SI, we added a figure (Supplementary Figure 1) illustrating the biomarker mass accumulation rates along with their concentrations. This clearly shows that, while amplified by spikes in sedimentation rates, the peaks in accumulation are already visible in the biomarker concentration records. Besides, we refer to our reply 2.2 above, where we elaborate why the records of accumulation rate are not related to the distance to the Amur river mouth.

Lines 95-96: it doesn't make sense here as B/A and PB are the two warmer periods before and after YD. Without evidence showing coastal erosion, it could also be the release of carbon from the drainage basin. Even so, coastal erosion and drainage basin inputs most likely work together.

The reviewer is right in pointing out that from the records of accumulation rate alone we cannot distinguish between coastal erosion and inland warming and erosion of

degrading inland permafrost. This is why we compare the biomarker accumulation rates with records of wetland development indicative of degradation of inland permafrost, and a reconstruction of the river discharge. Both records show a different timing with a maximum in the early Holocene temperature maximum. We do not neglect the possibility that during these phases, eroded pre-aged permafrost derived material is supplied by the river to our core sites (see lines 126 and following of the revised manuscript; no changes made).

Lines 96-97: The abrupt increase in biomarker accumulation, to me, is due to much higher sedimentation rates. I suspect biomarker concentrations don't change much through that time period. In addition, vegetation change at warming period may also contribute more HMW alkanes to the coast.

See our reply to comment re. line 87-88. It is also evident from the data of the depositional ages provided in table 1 that sedimentation rates did not spike strongly at times in question.

Line 101: During extreme events when sedimentation rates are much higher, the oxygen exposure time is shorter and the degradation is inhibited. I would like to see any degradation index through time to support the supposition of OC degradation.

Studying the degree of organic matter (OM) degradation in relation to the sedimentation rate and oxygen exposure time is beyond the scope of our study as we focus on mobilization of fossil carbon and not the process of OM degradation during transport; we refer to the abundant literature on this subject (e.g., see Vonk et al., 2010; Aller and Blair 2004).

Line 101: Why not transport to deep sea? What is the ice process in this region? Will that also affect the burial of terrestrial material?

We realize that the choice of wording was misleading here. We deleted "shallow" (line 111).

Line 114: the resolution of fig 2f is rather low, not many data points corresponding to the peaks in 2c/d. It is quite weak to make this statement.

The lower resolution of Figure 2f is due to the fact that the *Pediastrum* data are only available for core LV28-4, which has a lower accumulation rate during this time interval. We also added for reference the high-resolution record of monsoon precipitation in East Asia, which controls Amur River discharge. We disagree with reviewer#2 about our record not strongly supporting our statement – in the contrary, other authors have made similar observation in sediment cores from the Okhotsk Sea (e.g. Seki et al. 2012; Gorbarenko et al., 2014; Lembke-Jene et al., 2017).

No action taken.

Lines 117-120: Again, poor sampling for fig 2f, you cannot make such statement. The biomarker in fig 2f is derived from aquatic sources, while 2c/d are biomarkers for terrestrial sources in general, specifically soils and plants. They don't have to match with each other. But as I mentioned earlier, the evidence for coastal erosion is not well supported. Do you have any evidence showing where the Amur River has drain into the sea? Also, discharge and biomarker data are from two different cores which are quite far, so it is reasonable to have some timing difference.

As mentioned above, the *Pediastrum* accumulation rate is a well-established proxy for river discharge. While this is an accumulation rate which depends on local sediment accumulation, the second record presented as proxy for inland permafrost degradation, i.e., the Paq record in Fig. 2e, is based on an abundance ratio between different n-alkane

homologues and is thus independent on accumulation. Moreover, it was determined on yet another core (we cite Seki et al., 2012 for this records). The excellent agreement between these two records and opposing pattern to the records presented in Fig 2c/d is strong support for the validity of our statement.

The reviewer is correct in stating that absolute timing might differ slightly between river runoff as a function of climate variability and vegetation development. However, it has been shown that the response of vegetation to hydrologic changes is rapid, on the order of a few decades (e.g., Huguen et al., 2004). The difference in timing we discuss here, in contrast, is on the order of millennia, which is why we believe this is evidence for two independent processes, i.e., coastal erosion and inland warming.

With regards to the location of the Amur River mouth, we would like to refer again to our reply 2.2

Line 120: I doubt the application of algae biomarker in reconstructing discharge. To me, it is more representative of aquatic algae growth. This is a totally different term to discharge and they may vary independently.

The chlorococcal algae we used as evidence for freshwater outflow are commonly used as indicators for freshwater presence, also in limnic settings. A core-top study from the Arctic region demonstrates their applicability as proxies for river discharge (Matthiessen et al., 2000), and it is a frequently used proxy also across the timescales considered here (e.g., Ménot et al., 2006, Science). Commonly, *Pediastrum Spp.* does not survive salinities higher than 10 psu and is disintegrated relatively quickly in the water column, thus is not transported far after death. The presence of *Pediastrum Spp.* in our core thus indicates a significantly larger freshwater outflow compared to modern conditions, as the algae would not have been able to survive (let alone reproduce, indicating aquatic algae growth as presumed by reviewer#2) in even a brackish setting above the core site. Today, Amur outflow generates a low-salinity anomaly within the upper circa 20 m of the water column near our sites between 22-28 psu (Nishioka et al. 2014).

Line 126: What's the purpose of analyzing 2 cores with such far distance? Instead, by doing one core, you could increase the resolution of radiocarbon samples. I only see five and three on the figure3C. Where are the two other samples?

Please refer to our reply to comment 2.2 above, where we explain why using two cores strengthens our data base. Two data points are not displayed in Figure 3 for the sake of better visibility. We chose to display the data on a time scale from 4-18 kyrs BP. The data can be found in Table 1. We added this information to figure caption 3.

Lines 128-129: I don't think your data is showing this way. There are only eight data points, and only 2 after PB. For these 2 data points, one is comparable in age to the oldest pre PB, the other one is slightly lower than the youngest age pre PB. I understand CSRA requires a lot of work, but even so, such low sampling resolution could not support this statement.

Please see Tale 1 for the two additional CSRA dates as referred to in the text, which are not shown in this figure, because we concentrated on the deglacial; also see reply above (line 126). We added a clarification in the text (line 138-139).

Line 134: What is the yedoma distribution in this region?

Unfortunately, there are no available data on the past Yedoma distribution in this area. We follow Walter et al (2007) in their assumptions.

Line 135: Too many uses of vast in the MS.

We deleted “vast” and changed to large (lines 141 and 145)

Lines 145-146: The only evidence showing this is material from hinterland is through algae biomarker, which is suggestive of freshwater inputs. However, I think as a major conclusion of this manuscript, the discussion on terrestrial inputs through riverine transport is a bit of weak. You should find stronger evidence suggesting this is through riverine transport.

The reviewer did not consider that we provided an additional proxy record indicative of changes in the Amur river basin, i.e., the Paq record indicating wetland extent in the catchment. This record is independent (see above) and in excellent agreement with the discharge data.

Line 149: I think the results are not well discussed in terms of coastal erosion and hinterland inputs. It definitely makes sense that when the sea level was low, the sampling locations were way closer to the coast and therefore receive massive material from coastal erosion. While when sea level was higher, the sampling locations were away from the coast, and therefore were less affected by coastal erosion. However, I am surprised to see no discussion on water depth and its effect on bulk and biomarker signals.

Please see our reply 2.2. re. the expected effect of changes in the relative location of the river mouth to the core sites. Furthermore, we would like to stress again that at the pelagic water depths from which our cores were recovered (~700 m), effects of sea-level rise on the hydrological regime affecting the types of sediment deposited should be negligible.

Lines 151-156: Don't quite understand this part. Do you mean the material eroded now and 10 Ka ago is the same when considering the modern core top material is 18Ka, while the material 10Ka ago was 8 Ka when deposited? Mathematically it is correct, however, in reality, I doubt the truth of this statement. The permafrost landscape should have changed significantly with time considering the sea level, humidity, vegetation growth and human disturbance, etc. Thus such statement based just on carbon age is weak.

The reviewer is correct in assuming that overall, the permafrost landscapes were different between the last glacial maximum and today. However, in East Siberia today, we still see areas that are covered by relict deposits of the Last Glacial, which were unchanged but remained frozen since their formation tens of millennia ago. In our comparison, we refer to the age of material eroded precisely from these deposits and buried in marine sediments today. We use the analogy to underline that our assumption that the old terrestrial organic matter found in the deglacial sediments stems from Yedoma type deposits.

Lines 158-159: Don't see fatty acid data support this statement.

We changed the sentence to clarify: “During the period starting slightly before and encompassing MWP 1A, which was also characterized by slightly increasing river discharge, HMW *n*-alkanoic acids were about 6.5 kyrs in age at the time of deposition, and terrigenous biomarkers accumulated at high rates.” (lines 167-170)

Line 161: Not all Arctic, please be more specific.

We added information that these data are from the Lena river delta. (line 173)

Line 163: It should be a range based on Figure 3C.

Changed to “...of at least 5 kyrs in age....” to account for the fact that the age model uncertainty of LV28-4 allows the possibility that the terrigenous HMW *n*-alkanoic acids

at 751-753 cm core depth and dated to be >10 kyrs were deposited during the later YD. (line 174)

Line 166: The authors used a lot of modern analogs. However, in this case, the modern analog of coastal erosion only has carbon age of ~3-5 ka, much younger than the 10ka in this statement. I don't really believe it is through coastal erosion. Warming induced sea level rise could also stimulate export of terrestrial material through river transport. However, it is important to evaluate how largely the river affects this region first.

It appears that the reviewer misunderstood our analogies. Terrigenous organic matter ages of 3-5kyrs are related to inland-derived organic matter (Winterfeld et al., 2015; Vonk et al. 2014). Material derived from coastal erosion of Yedoma is much older (8-10 kyrs; Gustafsson et al., 2011; Vonk et al., 2012). We are, however, not sure which statement the reviewer refers to here, as in line 166 and following, no modern analogues are used.

No action taken.

Line 175: Paq is an empirical index and is affected by many factors, such as degradation, sources. I don't think it is the ideal proxy to be included in this case. Paq could also be affected by marine inputs, that is barely discussed in this manuscript. Terrestrial inputs, including nutrients, could stimulate marine primary production as well.

We refer to the cited literature for the proxy record we present (Seki et al., 2012) for a detailed discussion of the terrigenous organic matter sources supplied to the Okhotsk Sea. We also would like to stress again that our discharge record is in excellent agreement with the Paq records, although both records are completely independent. This is taken as evidence for the validity of both records and our interpretation of them.

No action taken.

Line 188: Throughout the manuscript, there is no proper discussion of permafrost carbon degradation. Although many cases have been published on modern systems, the statement of permafrost degradation is not a conclusion that can be reached in this study.

We use the term “degrading permafrost” in reference to the processes resulting in the disappearance of permafrost, which include thawing, thermokarst development, thaw slumping, and coastal erosion (now specifically stated in the revised manuscript, lines 52-55). We consistently use “degradation” in place of “thaw” throughout the revised manuscript to avoid confusion.

For clarity, we also modified the title to “Deglacial mobilization of pre-aged terrestrial carbon from **degrading** permafrost”.

The term “permafrost degradation”, however, does explicitly not refer to degradation (remineralization) of organic matter. In those cases where we discuss the microbial/photochemical remineralization (e.g., lines 224-227, 245-246, 301-305), the term “degradation” is not used.

Lines 191-193: add references indicating the glacial retreat in these areas.

During the last glacial maximum, East Siberia and Alaska (Beringia) was largely unglaciated and only small ice sheets covering the mountainous regions of Kamtchatka and the Kankaren Range (see e.g., Barr and Clark, 2011). This is why in these areas, thick organic rich glacial Yedoma deposits accumulated (cf. Strauss et al., 2017).

No action taken.

Lines 197-202: Is this only on Pleistocene time scale or from Pleistocene to the present?

We would like to refer to the reference (Strauss et al., 2017), who estimate the extent of Yedoma and its associated carbon pool for the last glacial maximum. We revised the description of our model assumptions (lines 200 and following paragraphs), which now clearly state which time spans we refer to.

Line 200: height to thickness.

The sentence to which this comment was referring to was deleted in the revised manuscript.

Line 210: Don't understand how this carbon could possibly be fully oxidized into CO₂ since it has been buried under sea during sea level rise with extra material accumulating on top of the yedoma since once underwater there are deposits of new material from terrestrial sources or marine sources. This carbon pool is likely permanently buried with minimal interaction to the atmosphere. However, I am surprised to see the whole model is based on this assumption, which to me is susceptible. See my major comment for details. To me, I think the majority role of permafrost thawing on atmospheric CO₂ level is through degradation of DOC during inland and riverine transport and soil respiration, however, the POC is less likely the main driver of atmospheric CO₂ levels.

We would like to refer to our replies above, i.e., 1.2 for the process of erosion during sea-level rise and 1.1 for the assumptions of carbon stocks and remineralization.

Line 213: I don't think it is doable this way by forcing 170PgC into these three events, especially when considering this number is generated using a simple box model.

Please see reply 1.1 for reviewer#1 and reply above for changes regarding this number and the model description. We would also like to stress here that the estimated amount of carbon release (i.e., 85 PgC in the revised manuscript) is not derived from a model but from recently published data (Brozius et al. 2012 for shelf area; Strauss et al., 2017 for Yedoma carbon stock and its change since the glacial; Vonk et al., 2012 for remineralization rate for Yedoma eroded along coasts; Lambeck et al., 2014 for sea-level rise) and the bathymetry of the Eastern Arctic and Bering Shelves. The total estimate of mobilized carbon is then distributed into three 200 year phases of pulsed release, and the respective size of these pulses is scaled based on our biomarker accumulation rate peaks.

We have edited the model description for clarity (lines 200 and following).

Line 221: Again, I am not sure if it is right to set the same time period of 200 for all three events. Check on the grammar of the sentence as well.

We refer to the reference cited here for the reasoning in the model setup. The duration of 200 years for the pulses was chosen to correspond to the periods of rapid CO₂ rise as recorded in the temporally highly resolved ice core records.

Line 222: I thought 14.6 Ka pins the start of carbon release in your case? I don't understand why 16.5 is included here. I can see the 16.5Ka peak is much smaller than the 11.5 Ka, so I don't really understand why the release rate is even higher for the 16.5 Ka.

There appears to be a misconception on the reviewer's side, and we would like to refer to our figure 4a, where the release rates are illustrated (thick red line). The release rate for the 16.5 kyr pulse is half that assumed for the later two phases. Overall, the model part of our manuscript was re-worded, and we changed some of the assumptions. We now trust that our reasoning is clearer (lines 200-248).

Lines 222-225: 1) I don't understand how this number is generated; 2) don't understand the mean of release here. Do you mean the released carbon goes to the sedimentary pool or the atmosphere? I don't think it is right to assume the release of carbon is proportional to the depositional rates of biomarkers while considering the sea level was not constant and sea level was rising. Earlier time when the sampling locations were closer to the coast, the sampling sites should have higher depositional rates of carbon, and later when the sea level was higher the sampling sites were farther away from the coast. Therefore, even if the drainage basin was under the same condition, the depositional rates of carbon should be lower.

We rephrased the model description: “In our model simulation, CO₂ release from permafrost, derived from the assumption that 66% of mobilized permafrost carbon was respired, was restricted to a time window of 200 years similar to a previous study³. Both the release length and the pinning of its onset to sea level changes was assumed to be identical for the two other events at 16.5 kyrs and 11.5 kyrs, while the annual release rates (0.17 or 0.09 PgC yr⁻¹) were derived from the total carbon amount that was assumed to be remineralized approximately scaled to the amplitudes of our biomarker MAR records. “(lines 238-250).

Secondly, we again refer to our reply 2.2, where we discuss why changes in sea-level did not influence the sedimentary setting of our core sites, making the record of terrigenous biomarker accumulation a reliable indicator of changes in their release from permafrost. As stated by the reviewer here, deglacial changes in the position of the river mouth and coast line relative to the core sites would be expected to result in a gradual decrease in accumulation of terrigenous material, which is not what we observe (distinct peaks in accumulation during the deglaciation, low values during the LGM and in the Holocene). So it is obvious that the observed record cannot be explained by the mechanism here proposed by the reviewer.

Lines 260-262: what about age model uncertainty?

As explained in the draft (discussion section) the onset of the terrestrial carbon release was based on U/Th dated ¹⁴C anomalies available for the 14.6 ka event to be synchronous with the onset of the abrupt sea level rise. There might be the possibility that the phasing of sea level rise and terrestrial carbon release might indeed be different in detail for the other events, such as the one at 16.5 ka. Furthermore, as pointed out by the reviewer, the possibility of age model uncertainties always exists, which would here refer to the synchronicity of abrupt sea level rise (marine records integrated in Lambeck et al 2014) and abrupt CO₂ rise (WAIS Divide ice core, Marcott et al., 2014). This problem of chronologies, however, is not only related to the event at 16.5 ka. In the revision we now briefly mention this.

Our changed paragraph can be found in lines 282-290 of the revised manuscript.

Line 265: what process?

In the previous sentence, we name “sea-level induced permafrost erosion”, which is what is meant here. We do not think that it would be good style to repeat this term again.

Lines 276-278: I would suspect the majority of permafrost derived carbon is reburied in sediments and this permafrost carbon is not the same source of carbon as discussed in the model. The permafrost you see in your sediment core is from the terrestrial land

above water, while the yedoma carbon pool you talked about is below water and is stable.

We refer to our reply to comment 1.2, where we discuss why Yedoma cannot be preserved below water.

Lines 278-279: This is the major factor affecting the estimates of permafrost carbon release back in time in many cases.

We agree that the carbon loss or remineralization rate is the crucial factor. We reworded this part to read:

“Furthermore, some of the permafrost-derived carbon is found in our sediment cores in the Okhotsk Sea, underlining the uncertainties associated with the estimated remineralization rate of 66%, which is potentially too high. The current range of estimates for carbon loss upon thaw varies from 2 to ~80% of the initial carbon stock^{35,36}.” (lines 301-305)

Lines 283-286: The marine side could not compensate the terrestrial carbon release. Marine takes CO₂ with minimal fractionation, therefore, if the CO₂ anomaly is due to terrestrial reasons, no matter what the marine side is doing, there should still be a significant shift in d13C.

The reviewer is referred to Crichton et al. (2016, *Nature Geosciences*), a study that examined the deglacial $\delta^{13}\text{C}$ record of CO₂ in context of the sources of carbon that lead to the atmospheric CO₂ increase. This study concludes that the atmospheric isotopic records is in agreement with permafrost carbon to be the source.

No action taken.

Line 301: the wetland formation in terms of CO₂ take-up is not discussed in this ms. If wetland could be such an important source of carbon in the sediment core in this study, then its impact on CO₂ level could not be ignored over geological time scales.

Line 302: It is rather a CO₂ sink than CH₄ source since it has net vegetation growth that takes up CO₂. But I do believe the abrupt CH₄ release could be due to wetland expansion globally.

We reply to both of the above comments in one paragraph: Based on our data, we cannot discuss the role of wetlands in terms of CO₂ uptake. There appears, however, to be another misunderstanding on the reviewer’s side: We never state that wetlands are an important source of carbon to the sediment core. We take their development as indication for permafrost degradation, which will also cause land sliding and erosion of permafrost; it is the latter two processes that might contribute permafrost derived carbon to the marine sediments.

No action taken.

Lines 307-309: But this does not explain the d13C.

Again, we would like to refer to Crichton et al., 2016 and Bauska et al, 2016, two studies we cite for their examination of the atmospheric $\delta^{13}\text{C}$ record during the deglaciation and the role permafrost degradation might have played in it. Both studies suggest that permafrost thawing and mobilization of carbon can explain parts of the atmospheric record.

Lines 309-311: Although it is possible, but high accumulation rates of pre-aged biomarkers does not suggest shelf erosion and flooding as two dominant processes.

This is our suggested interpretation. However, we acknowledge that the wording might have been misleading and therefore deleted “and flooding”, as we consider the sea-level rise induced erosion of deposits on the shelf the one dominant process. (line 338)

Lines 318-321: What about minimal changes in $\delta^{13}\text{C}$ values through that time period? Fig 1b: Apparently, back in time the Amur River drains into the ocean at a different location and how that would affect the consistency of your biomarker reconstruction and data interpretation is never discussed.

These points have been raised before and are discussed in several locations above (e.g. our replies re. lines 307-309)

Fig 2h: What are age control points? How do you get the age, forams or plants?

As described in the methods section, the age control points were derived from AMS ^{14}C dates of planktic foraminifera supplemented by ^{14}C dates of mollusks (see lines 396-399). We include a reference to Lembke-Jene et al. (2017) in the caption for figure 2h; the reference provided was an error for which we wish to apologize.

Reviewer #3 (Remarks to the Author):

Winterfeld et al present a powerful dataset that attempts to constrain the export of permafrost carbon after the last glacial maximum. Understanding the fate of Arctic permafrost in a future warmer Arctic is critical for estimating atmospheric CO_2 concentrations and using paleoclimate records through deglaciations may be one way to constrain the fate of permafrost carbon. This manuscript presents high quality compound specific radiocarbon data of alkanolic acids, carbon that was captured during photosynthesis on land and aged in situ before being transported to the ocean. Additionally, chemical biomarkers that characterize processes occurring on land were also presented. To me, the truly unique aspect of this study is the coupling the biomarker analysis and radiocarbon data with paleoclimate modelling to estimate how changes in the eroded permafrost could impact atmospheric CO_2 concentrations and radiocarbon composition. While the paper presents a novel approach to close this important gap in our understanding of Arctic carbon, as currently presented, this paper falls short of convincing this reviewer that the export of thawed permafrost carbon is responsible for changes in the deglaciation atmospheric CO_2 record.

We appreciate the reviewer’s assessment of our data set being a powerful constraint on the export of permafrost carbon during the last deglaciation.

To me, this manuscript lacked a unified narrative. Based on the title of the manuscript, the paper focused on the mobilization of aged permafrost carbon during deglaciation. Based on the abstract, the paper focused how sea level rise during deglaciation influences the delivery of aged permafrost carbon during deglaciation. The results presented data on the changes in the composition of terrigenous material exported to marine sediments during deglaciation. The discussion started by saying this exported material is responsible for changes in the abundance and isotopic composition of atmospheric CO_2 during deglaciation. While it is entirely possible that the erosion of old permafrost carbon could be oxidized to change atmospheric CO_2 , pulling the different

parts of the argument together in a more seamless fashion is required. I believe that with mindful editing, this paper can unify these at times seemingly disparate visions for the dataset.

We edited the abstract and introduction to improve the narrative. We also exchanged the word “thawing” by “degrading” in the title and throughout the text to avoid confusion.

It was not clear what the sources of organic carbon were to the cores studied. In the introduction, the manuscript states that there is high riverine sediment discharge (line 68) that gets accumulated in the ocean. Yet, later in the paper it is speculated that sea level rise caused coastal erosion of old permafrost carbon that was subsequently incorporated into the sediment cores (lines 93-100). Is that simply a correlation or is it causation? Does this mean that riverine sediments are not important?

We would like to refer to our reply 2.2, where we explain in detail the sediment provenance and the individual contributors of terrestrial material to the core sites.

Is there a way to estimate riverine based carbon from erosion based carbon? For example, would the composition of deep soil eroded from the coast have the same composition as permafrost carbon exported from the rivers? Are there mountains within the Amur River catchment area that could be responsible for the delivery of pre-aged carbon? At times during high discharge (Fig 2g), the age of the alkanolic acid also increases.

It is unfortunately not possible to distinguish unambiguously between terrestrial material that was remobilized by thawing inland and material that was eroded along the coasts, as we have to rely on the assumption of these deposits being similar to relict Pleistocene deposits found in northern East Asia today. To the best of our knowledge, there are no biomarkers or isotopic tracers available that allow distinguishing between inland and near-coast deposits (see studies by Vonk et al., 2010; Tesi et al., 2016, or Bröder et al., 2016, Keskitalo et al., 2017 for examples for the modern system).

Could this simply be an erosion signal from the mountains and not the mobilization of coastal permafrost that has thawed?

The Amur river basin is a large river basin, which includes mountainous regions in the headwaters and large lowlands. Several recent studies off large river basins have indicated that the organic geochemical signal recorded in marine sediments off these basins is to a substantial fraction controlled by material derived from the lowland regions (e.g., Galy et al., 2008; Kusch et al., 2010, Schefuss et al., 2011 *Nature*; Contreras-Rosales et al., 2014; Häggi et al., 2016; Freymond et al., 2018). Therefore, we do not consider deglaciation processes followed by erosion in the mountains likely causes for the records observed in our sediment cores.

It is critical that we constrain how much permafrost carbon can affect the atmospheric CO₂ concentration. Since aged permafrost carbon was clearly mobilized during deglaciation and the concentration and isotopic composition of atmospheric CO₂ changed during that time, it is plausible that the permafrost carbon could have contributed to the recorded changes in CO₂.

However, I have a few issues with this idea. The atmospheric concentration of CO₂ steadily increased during the Holocene, but it is also marked with periods of stability if not decline. For example, atmospheric [CO₂] rose during the HS1 yet stabilized during

B/A. Similarly, atmospheric [CO₂] rose during the YD but stabilized during the PB. How do these results explain the changes in the atmospheric CO₂?

We are not sure if we fully understand this comment. The reviewer is correct when stating that CO₂ rise during the transition from the last glacial to the Holocene was marked by periods of rapid rise and those with little change in CO₂ levels (as shown by Marcott et al., 2014, and also displayed on our Figure 3). We refer to the paper and our figure for the exact timing of these phases of rapid rise. It is particularly those phases of rapid rise during the onset of the B/A and the PB which we discuss as being related to permafrost carbon release, and our data indicate that this process might have played some role.

The ¹⁴C composition of atmospheric CO₂ does not appear to go through abrupt changes that coincide with the export of terrigenous material, so I am not convinced that the rapid delivery of thawed and pre-aged permafrost carbon will be oxidized and significantly contribute to the atmospheric CO₂ record.

We refer to Köhler et al., 2014 (*Nature Communications*) for details how $\Delta^{14}\text{C}$ changed during the phases of rapid atmospheric CO₂ rise. It is not correct to state that during the phases of rapid CO₂ rise, $\Delta^{14}\text{C}$ of atmospheric CO₂ remained unaffected. However, since $\Delta^{14}\text{C}$ is affected by several processes, including CO₂ release from aged reservoirs as well as other changes in carbon cycling and in its production rate controlled by cosmic radiation, its changes are not as drastic as those of CO₂. Moreover, both records are reconstructed from different archives, which explains a small degree of disagreement.

In order to illustrate which part of the $\Delta^{14}\text{C}$ change might be caused by permafrost carbon release, we include Figure 5b. In summary, the simulated changes in atmospheric ¹⁴C from degraded permafrost are so small (on the order of -10 ‰ per CO₂ peak) compared to the known changes in ¹⁴C production rate and the unexplained ¹⁴C residual (Fig 4c) that needs to be explained by carbon cycle changes that it can hardly be used to falsify if carbon release from permafrost might impact CO₂, as suggested by the reviewer here.

For that to happen would require many steps: export of pre-aged carbon (shown), full oxidation of said carbon (big assumption), all that oxidized CO₂ leaving the water column (assumed), thawed permafrost carbon in this region being representative of the rest of the world (assumed), and no other processes are leading to isotopic mixing of atmospheric ¹⁴CO₂ (assumed). While some permafrost carbon does get oxidized, as the authors state, the current estimates are between 2 and 88%. Mineral stabilization of organic matter protects a large portion of soil carbon from oxidation and this was overlooked.

It is correct that the assumption of full oxidation of mobilized organic matter (66% in the revised manuscript) is big. We deliberately chose a rather large remineralization rate as input in our model to derive an estimate of the maximum potential effect permafrost carbon mobilization could have. However, the reviewer overlooked that we are using a full carbon cycle model, which considers all relevant carbon cycle processes and in which most of the respired CO₂ remains in the ocean (with an airborne fraction of only 8 % on a 8-kyr time scale, (Supplementary Figure 3a)).

We furthermore assume that our records indicating that sea-level rise is a strong control on carbon mobilization from permafrost are representative for this process being active across large regional scales. The shelf areas of the Bering Sea and of East Siberia, which are among the largest shelf areas in the world, were covered by extensive permafrost deposits (likely to a substantial fraction ice- and carbon rich Yedoma, see Walter et al.,

2007, or Strauss et al., 2017). These shelf areas have been flooded during the last deglaciation, and there are several previously published records showing the effect of sea-level rise on accumulation of terrigenous material in marine sediments (e.g., Bauch et al., 2001; Fahl and Stein, 2012, Tesi et al., 2016, *Nature Communications*; Keskitalo et al., 2017). Therefore, we consider this assumption reasonable.

The isotopic composition of the atmosphere, in particular the $\Delta^{14}\text{C}$ signature, is the result of several processes, including release of ^{14}C -depleted CO_2 from the ocean and, potentially from land. This is fully acknowledged in our model. We are examining only the change in $\Delta^{14}\text{C}$ that currently remains unexplained by the other processes (see our figure 5b). We also added a sentence clarifying the extent to which permafrost thaw might have impacted atmospheric $\Delta^{14}\text{C}$ (lines 343-349).

Again, as stated before, we are fully aware that the estimates of remineralization rates vary over a wide range. Protection by mineral association is one of the possible mechanisms, others include rapid burial in marine sediments. Within the scope of this study, we cannot assess which of these protective mechanisms affected the preservation of terrestrial organic matter in our sediment, but we would like to stress that we are fully aware of organic matter protection.

Finally, mechanisms for the oxidation of aged permafrost were missing. Do the authors propose that the carbon undergoes microbial oxidation? Or would photochemical processing also change this material - albeit under limited light conditions due to high sediment load and latitude? The mechanism for this oxidation process could influence the ^{13}C composition of the oxidized carbon.

Mechanisms for the oxidation of aged permafrost carbon today include microbial oxidation of organic matter in soils, peat bogs, ponds, and lakes, microbial and photochemical oxidation of dissolved organic matter in meltwater creeks, streams, and rivers, and likely microbial oxidation of particulate organic matter (see, e.g., Knoblauch et al., 2018, Mann et al., 2015, Sanchez-Garcia et al., 2014). These processes known for the modern system are described in the manuscript (lines 114-119). We infer that similar processes were active in the past, which we also mention in the manuscript (line 114). Going further into detail regarding the mechanisms of oxidation, however, would be purely speculative at this stage, which is why we refrained from it.

As the authors note, the atmospheric $^{13}\text{C}\text{CO}_2$ recorded in ice cores illustrate changes in the ^{13}C of CO_2 during the deglaciation and the $^{13}\text{C}\text{CO}_2$ records do not reflect a release in terrestrial carbon. The paragraph acknowledging the lack of agreement with the ^{13}C record (lines 280-289)

We would like to stress that it is controversially debated in the literature whether the ^{13}C signature of CO_2 is consistent with terrestrial sources of the CO_2 release (see lines 306-310 of the revised manuscript).

then made me re-examine and question the agreement between the $^{14}\text{C}\text{CO}_2$ record and the exported pre-aged permafrost carbon. But it also highlighted an additional uncertainty for me: what is the $^{13}\text{C}\text{CO}_2$ composition of photo-oxidized organic matter? CO_2 evolved from systems that are not carbon constrained undergo biological fractionation of ^{13}C , which leads to a depleted $^{13}\text{C}\text{CO}_2$.

The pathway of oxidation of carbon released from degrading permafrost is not clear, nor may it be uniform. It likely includes both microbial respiration and photo-oxidation. At this point, it would be highly speculative to include a discussion on the possible effects of these remineralization processes on the stable carbon isotopic signature. In a first approximation, we would assume full oxidation of organic matter by either of these processes, leading to no isotopic fractionation in the resulting CO_2 gas.

By my reading of Figure 3, the ^{14}C values of the alkanolic acids in the two cores did not completely tell the same story. The LV28-4-4 samples were more pre-aged at times that didn't coincide with thawing the same as the SO178-13-6 samples (Figure 3). What could explain this difference in the amount of pre-aged material? Was the biomarker composition (or bulk composition - C/N, $\delta^{13}\text{C}_{\text{org}}$, etc) similar between the two cores? Could this disparity between the two cores be related to the sample size of the ^{14}C samples? I noticed that the LV28-4-4 samples were smaller and could therefore contained more exogenous carbon?

All of our compound-specific radiocarbon results were corrected for process blanks/contributions of exogenous carbon. These corrections take the size of the samples into account. We refer to the details presented in the supplement. Moreover, the radiocarbon composition of our blank was estimated to be $F^{14}\text{C}$ 0.6776, which would bias our relatively old biomarker ages toward younger values. We therefore do not think that higher amounts of blanks explain the disparity between the ^{14}C values. Since core LV28-4 has a substantially lower sedimentation rate in the deglaciation, its age model uncertainty is larger, reflected in the error bars displayed on figure 3. We therefore did not want to over-interpret the rather old HMW *n*-alkanoic acid age at the end of the YD found in this core, as age-model uncertainty (as reflected in the error bars displayed on the figure) allow placement of this core depth in the Pre-Boreal (during MWP 1b) as well, which in the other core is characterized by old (~8 kyrs) HMW *n*-alkanoic acids. We changed the sentence to “Synchronous to the phase of lowest deglacial sea-level rise, HMW *n*-alkanoic acids of at least 5 kyrs in age were deposited during the Younger Dryas cold period.” (line 173-175).

Other notes:

Figure 1: I found these maps confusing. The current shoreline was not evident and due to the greyscale contrast of the seafloor, the nearby subduction was a focal point. Please edit these maps to clearly highlight the current sea shore and the study area.

In both panels, the current shoreline is marked (solid line in panel a illustrating the present situation, dashed line in panel b showing the glacial extent of permafrost). Perhaps the resolution of the version the reviewer examined was insufficient. We changed the figure caption accordingly.

Table 1 + Supplementary: I appreciated the efforts in generating accurate and precise radiocarbon measurements. This does not always happen. Since the R2 was 'ignored' for the C30 (^{14}C free material) how was the uncertainty of the modern contamination estimated?

Does an uncertainty of blank of 0.08 $\mu\text{g C}$ seem reasonable based on other studies that have assessed ^{14}C contamination in CSRA samples? This seems small to me - as does the uncertainty of the CSRA samples given in Table 1.

It is correct that the estimated uncertainty of the blank of 0.08 $\mu\text{g C}$ is lower than given in other references, e.g., in Tao et al., 2015. However, in our approach we follow the blank assessment study by Shah and Pearson, 2007, and here only the regression coefficient is used to estimate the uncertainty. As for the ^{14}C free material, we could not derive a regression coefficient, as insufficient data points were available, we were left with using only one coefficient. In follow up studies, we will make sure to include additional data for the assessment of blank sizes, but unfortunately this cannot be done for this data set. However, as the ages we interpret to indicate thawing of permafrost are several thousand years higher than those found in the late Holocene under permafrost

free conditions, we are confident that this apparently low estimate of uncertainty in the size of the blank does not impair the validity of our statements.

References:

- Aller, R.C. and Blair, N. (2004) Early diagenetic remineralization of sedimentary organic C in the Gulf of Papua deltaic complex (Papua New Guinea): Net loss of terrestrial C and diagenetic fractionation of C isotopes. *Geochimica et Cosmochimica Acta* 68, 1815-1825.
- Barnhart, K.R., Anderson, R.S., Overeem, I., Wobus, C., Clow, G.D. and Urban, F.E. (2014) Modeling erosion of ice-rich permafrost bluffs along the Alaskan Beaufort Sea coast. *Journal of Geophysical Research: Earth Surface* 119, 1155-1179.
- Barr, I.D. and Clark, C.D. (2012) Late Quaternary glaciations in Far NE Russia; combining moraines, topography and chronology to assess regional and global glaciation synchrony. *Quaternary Science Reviews* 53, 72-87.
- Bauch, H.A., Mueller-Lupp, T., Taldenkova, E., Spielhagen, R.F., Kassens, H., Grootes, P.M., Thiede, J., Heinemeier, J. and Petryashov, V.V. (2001) Chronology of the Holocene transgression at the North Siberian margin. *Global and Planetary Change* 31, 125-139.
- Bergamaschi, B.A., Tsamakis, E., Keil, R.G., Eglinton, T.I., Montluçon, D.B. and Hedges, J.I. (1997) The effect of grain size and surface area on organic matter, lignin and carbohydrate concentration, and molecular compositions in Peru Margin sediments. *Geochimica et Cosmochimica Acta* 61, 1247-1260.
- Brosius, L.S., Walter Anthony, K.M., Grosse, G., Chanton, J.P., Farquharson, L.M., Overduin, P.P. and Meyer, H. (2012) Using the deuterium isotope composition of permafrost meltwater to constrain thermokarst lake contributions to atmospheric CH₄ during the last deglaciation. *Journal of Geophysical Research: Biogeosciences* 117.
- Contreras-Rosales, L.A., Schefuß, E., Meyer, V., Palamenghi, L., Lückge, A. and Jennerjahn, T.C. (2016) Origin and fate of sedimentary organic matter in the northern Bay of Bengal during the last 18ka. *Global and Planetary Change* 146, 53-66.
- Crichton, K.A., Bouttes, N., Roche, D.M., Chappellaz, J. and Krinner, G. (2016) Permafrost carbon as a missing link to explain CO₂ changes during the last deglaciation. *Nature Geosci* 9, 683-686.
- Dupeyrat, L., Costard, F., Randriamazaoro, R., Gailhardis, E., Gautier, E. and Fedorov, A. (2011) Effects of Ice Content on the Thermal Erosion of Permafrost: Implications for Coastal and Fluvial Erosion. *Permafrost and Periglacial Processes* 22, 179-187.
- Eglinton, T.I., Eglinton, G., Dupont, L.M., Sholkovitz, E.R., Montluçon, D. and Reddy, C.M. (2002) Composition, age, and provenance of organic matter in NW African dust over the Atlantic Ocean. *Geochemistry, Geophysics, Geosystems*, G3 3, 10.1029/2001GC000269.
- Freymond, C.V., Kündig, N., Stark, C., Peterse, F., Bugge, B., Lupker, M., Plötze, M., Blattmann, T.M., Filip, F., Giosan, L. and Eglinton, T.I. (2018) Evolution of biomolecular loadings along a major river system. *Geochimica et Cosmochimica Acta* 223, 389-404.
- Galy, V., François, L., France-Lanord, C., Faure, P., Kudrass, H., Pailhol, F. and Singh, S.K. (2008) C₄ plants decline in the Himalayan basin since the Last Glacial Maximum. *Quaternary Science Reviews* 27, 1396-1409.
- Galy, V., Peucker-Ehrenbrink, B. and Eglinton, T. (2015) Global carbon export from the terrestrial biosphere controlled by erosion. *Nature* 521, 204.
- Gorbarenko, S.A., Artemova, A.V., Goldberg, E.L. and Vasilenko, Y.P. (2014) The response of the Okhotsk Sea environment to the orbital-millennium global climate changes during the Last Glacial Maximum, deglaciation and Holocene. *Global and Planetary Change* 116, 76-90.
- Gorbarenko, S.A., Psheneva, O.Y., Artemova, A.V., Matul', A.G., Tiedemann, R. and Nürnberg, D. (2010) Paleoenvironment changes in the NW Okhotsk Sea for the last 18kyr determined with micropaleontological, geochemical, and lithological data. *Deep Sea Research Part I: Oceanographic Research Papers* 57, 797-811.
- Günther, F., Overduin, P.P., Sandakov, A.V., Grosse, G. and Grigoriev, M. (2013) Short- and long-term thermo-erosion of ice-rich permafrost coasts in the Laptev Sea region. *Biogeosciences* 10, 4297-4318.
- Gustafsson, Ö., van Dongen, B.E., Vonk, J.E., Dudarev, O.V. and Semiletov, I.P. (2011) Widespread release of old carbon across the Siberian Arctic echoed by its large rivers. *Biogeosciences* 8, 1737-1743.
- Häggi, C., Sawakuchi, A.O., Chiessi, C.M., Mülitz, S., Mollenhauer, G., Sawakuchi, H.O., Baker, P.A., Zabel, M. and Schefuß, E. (2016) Origin, transport and deposition of leaf-wax biomarkers in the Amazon Basin and the adjacent Atlantic. *Geochimica et Cosmochimica Acta* 192, 149-165.
- Hoque, M.A. and Pollard, W.H. (2016) Stability of permafrost dominated coastal cliffs in the Arctic. *Polar*

- Science 10, 79-88.
- Hugelius, G., Strauss, J., Zubrzycki, S., Harden, J.W., Schuur, E.A.G., Ping, C.L., Schirrmeister, L., Grosse, G., Michaelson, G.J., Koven, C.D., O'Donnell, J.A., Elberling, B., Mishra, U., Camill, P., Yu, Z., Palmtag, J. and Kuhry, P. (2014) Estimated stocks of circumpolar permafrost carbon with quantified uncertainty ranges and identified data gaps. *Biogeosciences* 11, 6573-6593.
- Hughen, K., Eglinton, T.I., Xu, L. and Makou, M. (2004) Abrupt tropical vegetation response to rapid climate change. *Science* 304, 1955-1959.
- Keskitalo, K., Tesi, T., Bröder, L., Andersson, A., Pearce, C., Sköld, M., Semiletov, I., Dudarev, O. and Gustafsson, Ö. (2017) Sources and characteristics of terrestrial carbon in Holocene-scale sediments of the East Siberian Sea. *Climate of the Past* 13, 1213-1226.
- Knoblauch, C., Beer, C., Liebner, S., Grigoriev, M.N. and Pfeiffer, E.-M. (2018) Methane production as key to the greenhouse gas budget of thawing permafrost. *Nature Climate Change* 8, 309-312.
- Köhler, P., Knorr, G. and Bard, E. (2014) Permafrost thawing as a possible source of abrupt carbon release at the onset of the Bølling/Allerød. *Nature Communications* 5, 5520.
- Kusch, S., Eglinton, T.I., Mix, A. and Mollenhauer, G. (2010) Timescales of lateral sediment transport in the Panama Basin as revealed by compound-specific radiocarbon ages of alkenones. *Earth and Planetary Science Letters* 290, 340-350.
- Lambeck, K., Rouby, H., Purcell, A., Sun, Y. and Sambridge, M. (2014) Sea level and global ice volumes from the Last Glacial Maximum to the Holocene. *Proceedings of the National Academy of Sciences* 111, 15296-15303.
- Lantuit, H., Atkinson, D., Paul Overduin, P., Grigoriev, M., Rachold, V., Grosse, G. and Hubberten, H.-W. (2011) Coastal erosion dynamics on the permafrost-dominated Bykovsky Peninsula, north Siberia, 1951–2006. *Polar Research* 30, 7341.
- Lantuit, H., Overduin, P.P., Couture, N., Wetterich, S., Aré, F., Atkinson, D., Brown, J., Cherkashov, G., Drozdov, D., Forbes, D.L., Graves-Gaylord, A., Grigoriev, M.N., Hubberten, H.W., Jordan, J., Jorgenson, T., Odegard, R.S., Ogorodov, S., Pollard, W.H., Rachold, V., Sedenko, S., Solomon, S., Steenhuisen, F., Streletskaia, I. and Vasiliev, A. (2012) The Arctic Coastal Dynamics Database: A New Classification Scheme and Statistics on Arctic Permafrost Coastlines. *Estuaries and Coasts* 35, 383-400.
- Lembke-Jene, L., Tiedemann, R., Nürnberg, D., Kokfelt, U., Kozdon, R., Max, L., Röhl, U. and Gorbarenko, S.A. (2017) Deglacial variability in Okhotsk Sea Intermediate Water ventilation and biogeochemistry: Implications for North Pacific nutrient supply and productivity. *Quaternary Science Reviews* 160, 116-137.
- Mann, P.J., Eglinton, T.I., McIntyre, C.P., Zimov, N., Davydova, A., Vonk, J.E., Holmes, R.M. and Spencer, R.G.M. (2015) Utilization of ancient permafrost carbon in headwaters of Arctic fluvial networks. *Nature Communications* 6, 7856.
- Marcott, S.A., Bauska, T.K., Buizert, C., Steig, E.J., Rosen, J.L., Cuffey, K.M., Fudge, T.J., Severinghaus, J.P., Ahn, J., Kalk, M.L., McConnell, J.R., Sowers, T., Taylor, K.C., White, J.W.C. and Brook, E.J. (2014) Centennial-scale changes in the global carbon cycle during the last deglaciation. *Nature* 514, 616.
- Matthiessen, J., Kunz-Pirrung, M. and Mudie, P.J. (2000) Freshwater chlorophycean algae in recent marine sediments of the Beaufort, Laptev and Kara Seas (Arctic Ocean) as indicators of river runoff. *International Journal of Earth Sciences* 89, 470-485.
- Ménot, G., Bard, E., Rostek, F., Weijers, J.W.H., Hopmans, E.C., Schouten, S. and Sinninghe Damsté, J.S. (2006) Early reactivation of European rivers during the last deglaciation. *Science* 313, 1623-1625.
- Mollenhauer, G. and Eglinton, T.I. (2007) Diagenetic and sedimentological controls on the composition of organic matter preserved in California Borderland Basin sediments. *Limnology and Oceanography* 52, 558-576.
- Mollenhauer, G., Eglinton, T.I., Ohkouchi, N., Schneider, R.R., Müller, P.J., Grootes, P.M. and Rullkötter, J. (2003) Asynchronous alkenone and foraminifera records from the Benguela Upwelling System. *Geochimica et Cosmochimica Acta* 67, 2157-2171.
- Mollenhauer, G., Inthorn, M., Vogt, T., Zabel, M., Sinninghe Damsté, J.S. and Eglinton, T.I. (2007) Aging of marine organic matter during cross-shelf lateral transport in the Benguela upwelling system revealed by compound-specific radiocarbon dating. *Geochemistry, Geophysics, Geosystems*, G3 8, Q09004.
- Nishioka, J., Nakatsuka, T., Ono, K., Volkov, Y.N., Scherbinin, A. and Shiraiwa, T. (2014) Quantitative evaluation of iron transport processes in the Sea of Okhotsk. *Progress in Oceanography* 126, 180-193.
- Ohkouchi, N., Kawamura, K., Kawahata, H. and Taira, A. (1997) Latitudinal distributions of terrestrial biomarkers in the sediments from the Central Pacific. *Geochimica et Cosmochimica Acta* 61, 1911-

- 1918.
- Ohshima, K.I., Simizu, D., Itoh, M., Mizuta, G., Fukamachi, Y., Riser, S.C. and Wakatsuchi, M. (2004) Sverdrup Balance and the Cyclonic Gyre in the Sea of Okhotsk. *Journal of Physical Oceanography* 34, 513-525.
- Ohshima, K.I. and Martin, S. (2004) Introduction to special section: Oceanography of the Okhotsk Sea. *Journal of Geophysical Research: Oceans* 109.
- Pearson, A. and Eglinton, T.I. (2000) The origin of n-alkanes in Santa Monica Basin surface sediment: a model based on compound-specific D¹⁴C and d¹³C. *Organic Geochemistry* 31, 1103-1116.
- Sanchez-Garcia, L., Vonk, J.E., Charkin, A., Kosmach, D., Dudarev, O., Semiletov, I. and Gustafsson, Ö. (2014) Characterisation of Three Regimes of Collapsing Arctic Ice Complex Deposits on the SE Laptev Sea Coast using Biomarkers and Dual Carbon Isotopes. *Permafrost and Periglacial Processes* 25, 172-183.
- Schefuß, E., Eglinton, T.I., Spencer-Jones, C.L., Rullkötter, J., De Pol-Holz, R., Talbot, H.M., Grootes, P.M. and Schneider, R.R. (2016) Hydrologic control of carbon cycling and aged carbon discharge in the Congo River basin. *Nature Geoscience* 9, 687.
- Schefuß, E., Kuhlmann, H., Mollenhauer, G., Prange, M. and Patzold, J. (2011) Forcing of wet phases in southeast Africa over the past 17,000 years. *Nature* 480, 509-512.
- Seki, O., Harada, N., Sato, M., Kawamura, K., Ijiri, A. and Nakatsuka, T. (2012) Assessment for paleoclimatic utility of terrestrial biomarker records in the Okhotsk Sea sediments. *Deep Sea Research Part II: Topical Studies in Oceanography* 61-64, 85-92.
- Seki, O., Mikami, Y., Nagao, S., Bendle, J.A., Nakatsuka, T., Kim, V.I., Shesterkin, V.P., Makinov, A.N., Fukushima, M., Moossen, H.M. and Schouten, S. (2014) Lignin phenols and BIT index distributions in the Amur River and the Sea of Okhotsk: Implications for the source and transport of particulate terrestrial organic matter to the ocean. *Progress in Oceanography* 126, 146-154.
- Seki, O., Yoshikawa, C., Nakatsuka, T., Kawamura, K. and Wakatsuchi, M. (2006) Fluxes, source and transport of organic matter in the western Sea of Okhotsk: Stable carbon isotopic ratios of n-alkanes and total organic carbon. *Deep Sea Research Part I: Oceanographic Research Papers* 53, 253-270.
- Seki, O., Kawamura, K., Nakatsuka, T., Ohnishi, K., Ikehara, M. and Wakatsuchi, M. (2003) Sediment core profiles of long-chain n-alkanes in the Sea of Okhotsk: Enhanced transport of terrestrial organic matter from the last deglaciation to the early Holocene. *Geophysical Research Letters* 30, 1-1-1-4.
- Shah, S.R. and Pearson, A. (2007) Ultra-microscale (5-25 µg C) analysis of individual lipids by ¹⁴C AMS: Assessment and correction for sample processing blanks. *Radiocarbon* 49, 69-82.
- Shakhova, N., Semiletov, I., Gustafsson, O., Sergienko, V., Lobkovsky, L., Dudarev, O., Tumskey, V., Grigoriev, M., Mazurov, A., Salyuk, A., Ananiev, R., Koshurnikov, A., Kosmach, D., Charkin, A., Dmitrevsky, N., Karnaukh, V., Gunar, A., Meluzov, A. and Chernykh, D. (2017) Current rates and mechanisms of subsea permafrost degradation in the East Siberian Arctic Shelf. *Nature Communications* 8, 15872.
- Shakhova, N., Semiletov, I., Salyuk, A., Yusupov, V., Kosmach, D. and Gustafsson, Ö. (2010) Extensive Methane Venting to the Atmosphere from Sediments of the East Siberian Arctic Shelf. *Science* 327, 1246.
- Shcherbina, A.Y., Talley, L.D. and Rudnick, D.L. (2004) Dense water formation on the northwestern shelf of the Okhotsk Sea: 1. Direct observations of brine rejection. *Journal of Geophysical Research: Oceans* 109.
- Strauss, J., Schirrmeister, L., Grosse, G., Fortier, D., Hugelius, G., Knoblauch, C., Romanovsky, V., Schädel, C., Schneider von Deimling, T., Schuur, E.A.G., Shmelev, D., Ulrich, M. and Veremeeva, A. (2017) Deep Yedoma permafrost: A synthesis of depositional characteristics and carbon vulnerability. *Earth-Science Reviews* 172, 75-86.
- Strauss, J., Schirrmeister, L., Grosse, G., Wetterich, S., Ulrich, M., Herzschuh, U. and Hubberten, H.-W. (2013) The deep permafrost carbon pool of the Yedoma region in Siberia and Alaska. *Geophysical Research Letters* 40, 6165-6170.
- Tao, S., Eglinton, T.I., Montluçon, D.B., McIntyre, C. and Zhao, M. (2015) Pre-aged soil organic carbon as a major component of the Yellow River suspended load: Regional significance and global relevance. *Earth and Planetary Science Letters* 414, 77-86.
- Ternois, Y., Kawamura, K., Keigwin, L., Ohkouchi, N. and Nakatsuka, T. (2001) A biomarker approach for assessing marine and terrigenous inputs to the sediments of Sea of Okhotsk for the last 27,000 years. *Geochimica et Cosmochimica Acta* 65, 791-802.
- Tesi, T., Muschitiello, F., Smittenberg, R.H., Jakobsson, M., Vonk, J.E., Hill, P., Andersson, A., Kirchner, N., Noormets, R., Dudarev, O., Semiletov, I. and Gustafsson, Ö. (2016) Massive remobilization of permafrost carbon during post-glacial warming. *Nature Communications* 7, 13653.

- Tesi, T., Semiletov, I., Dudarev, O., Andersson, A. and Gustafsson, Ö. (2016) Matrix association effects on hydrodynamic sorting and degradation of terrestrial organic matter during cross-shelf transport in the Laptev and East Siberian shelf seas. *Journal of Geophysical Research: Biogeosciences* 121, 731-752.
- Vonk, J.E., Sanchez-Garcia, L., van Dongen, B.E., Alling, V., Kosmach, D., Charkin, A., Semiletov, I.P., Dudarev, O.V., Shakhova, N., Roos, P., Eglinton, T.I., Andersson, A. and Gustafsson, O. (2012) Activation of old carbon by erosion of coastal and subsea permafrost in Arctic Siberia. *Nature* 489, 137-140.
- Vonk, J.E., Semiletov, I.P., Dudarev, O.V., Eglinton, T.I., Andersson, A., Shakhova, N., Charkin, A., Heim, B. and Gustafsson, Ö. (2014) Preferential burial of permafrost-derived organic carbon in Siberian-Arctic shelf waters. *Journal of Geophysical Research: Oceans* 119, 8410-8421.
- Vonk, J.E., van Dongen, B.E. and Gustafsson, Ö. (2010) Selective preservation of old organic carbon fluvially released from sub-Arctic soils. *Geophys. Res. Lett.* 37, L11605.
- Walter, K.M., Edwards, M.E., Grosse, G., Zimov, S.A. and Chapin, F.S. (2007) Thermokarst Lakes as a Source of Atmospheric CH₄ During the Last Deglaciation. *Science* 318, 633-636.
- Winterfeld, M., Laepple, T. and Mollenhauer, G. (2015) Characterization of particulate organic matter in the Lena River delta and adjacent nearshore zone, NE Siberia – Part I: Radiocarbon inventories. *Biogeosciences* 12, 3769-3788.
- Winterfeld, M., Schirrmeister, L., Grigoriev, M.N., Kunitsky, V.V., Andreev, A., Murray, A. and Overduin, P.P. (2011) Coastal permafrost landscape development since the Late Pleistocene in the western Laptev Sea, Siberia. *Boreas* 40, 697-713.

Reviewers' comments:

Reviewer #1 (Remarks to the Author):

Review of: Deglacial mobilization of pre-aged terrestrial carbon from degrading permafrost

Winterfeld et al.

Overall Evaluation:

I am generally satisfied with the revisions made to the original manuscript and recommend publication with only a few minor revisions.

Minor Revisions:

Line 29: Change 'here presently' to 'presently here'

Line 68 to 69: Change 'here presently' to 'presently here'

Line 71: After 'Okhotsk Sea' add 'of the Pacific Ocean', to better orient the reader.

Line 69: Change 'off' to 'off of'

Line 248: Delete 'each' change to '34 PgC at both 11.5 and ...'

Figure 2: Change ka to kyrs

Figure 3: Change ka to kyrs

Reviewer #2 (Remarks to the Author):

I am impressed that authors have done tremendous amount of work in dealing with the comments raised by me and other two reviewers. Even though I do not see many revisions being made in the main text from previous manuscript, they have done decent job in further clarifying descriptions in the manuscripts and rebutting comments from me. I am still not convinced by the modeling part, however, I do not think that would lower the contribution of this manuscript to the field of permafrost study. I would like to see this manuscript being published. --Xingqian

Reviewer #3 (Remarks to the Author):

Reviewer #3 (original review): It was not clear what the sources of organic carbon were to the cores studied. In the introduction, the manuscript states that there is high riverine sediment discharge (line 68) that gets accumulated in the ocean. Yet, later in the paper it is speculated that sea level rise caused coastal erosion of old permafrost carbon that was subsequently incorporated into the sediment cores (lines 93-100). Is that simply a correlation or is it causation? Does this mean that riverine sediments are not important?

Authors' response: We would like to refer to our reply 2.2, where we explain in detail the sediment provenance and the individual contributors of terrestrial material to the core sites.

Reviewer #3 (Reviewing the revision): After reading through the revised manuscript and the authors' reply to 2.2, the justification of sea level rise versus riverine material is still not clear to me. While I am only one reader, if I suspect if I am having this confusion, the broad audience of this journal will also struggle with this distinction of the sources of material to the sediment. In the revised results, (lines 103-106) it is stated that the highest rates of sediment accumulation coincide with melt water pulses suggesting that coastal erosion was the main cause of the carbon mobilization. The next paragraph talks about how riverine contributions play an important role. But the rest of the manuscript focuses solely on the coast erosion.

Below are a few suggestions on how you may be able to persuade reader of this assertion. 1) Consider mentioning the magnitude of sea level rise during this time period here and not waiting until line 218 to share this information will help the readers. Not everyone reading this paper will be familiar with that information. 2) I suggest you make the timing difference between the discharge and the accumulation rate clearer. Currently its written that there is a difference in timing (line 127). But I think it could be clearer for your readers if you say that the discharge happened after the peak in accumulation. 3) A back of the envelope calculation of how much carbon could be mobilized by 80 meters of sea level rise may be useful for readers. For example, if sea level rise rose by 80 meters, it flooded X amount of land and therefore potentially mobilized Z amount of C. And then if this amount of C was compared to contemporary riverine export would be helpful, as it would put the scale of the mobilized C into perspective.

Reviewer #3 (original review): Figure 1: I found these maps confusing. The current shoreline was not evident and due to the greyscale contrast of the seafloor, the nearby subduction was a focal point. Please edit these maps to clearly highlight the current sea shore and the study area.

Authors' response: In both panels, the current shoreline is marked (solid line in panel a illustrating the present situation, dashed line in panel b showing the glacial extent of permafrost). Perhaps the resolution of the version the reviewer examined was insufficient. We changed the figure caption accordingly.

Reviewer #3 (Reviewing the revision): When I printed the revised manuscript in black and white, I still found the maps hard to read. It is likely that other readers of this paper will also find it difficult to read the maps. Please try looking at them again in B&W and consider the broad audience who may not be used to looking at maps in this region.

Three reviewers commented on the revised version of our original submission. While Reviewer 1 and 2 seemed to be overall satisfied with our replies to their previous comments and only had minor points (Reviewer 1), Reviewer 3 made some more detailed suggestions. Below please find the Reviewers' comments in black and our replies in blue.

Reviewers' comments:

Reviewer #1 (Remarks to the Author):

Review of: Deglacial mobilization of pre-aged terrestrial carbon from degrading permafrost

Winterfeld et al.

Overall Evaluation:

I am generally satisfied with the revisions made to the original manuscript and recommend publication with only a few minor revisions.

Minor Revisions:

Line 29: Change 'here presently' to 'presently here'

Line 68 to 69: Change 'here presently' to 'presently here'

Reply: the reviewer apparently misread our text, as it states “we here present” instead of “here presently” in both sentences indicated above. Therefore, we did not make any changes as we believe that our wording is correct.

Line 71: After 'Okhotsk Sea' add 'of the Pacific Ocean', to better orient the reader.

Reply: We changed the sentence to “...deposited in marine sediments off the mouth of the Amur River in the Okhotsk Sea, a marginal sea of the North Pacific Ocean.”

Line 69: Change 'off' to 'off of'

done

Line 248: Delete 'each' change to '34 PgC at both 11.5 and ...'

done

Figure 2: Change ka to kyrs

done

Figure 3: Change ka to kyrs

done

Reviewer #2 (Remarks to the Author):

I am impressed that authors have done tremendous amount of work in dealing with the comments raised by me and other two reviewers. Even though I do not see many revisions being made in the main text from previous manuscript, they have done decent job in further clarifying descriptions in the manuscripts and rebutting comments from me. I am still not convinced by the modeling part, however, I do not think that would lower the contribution of this manuscript to the field of permafrost study. I would like to see this manuscript being published. --Xingqian

We thank Xingqian for this positive feedback.

Reviewer #3 (Remarks to the Author):

Reviewer #3 (original review): It was not clear what the sources of organic carbon were to the cores studied. In the introduction, the manuscript states that there is high riverine sediment discharge (line 68) that gets accumulated in the ocean. Yet, later in the paper it is speculated that sea level rise caused coastal erosion of old permafrost carbon that was subsequently incorporated into the sediment cores (lines 93-100). Is that simply a correlation or is it causation? Does this mean that riverine sediments are not important?

Authors' response: We would like to refer to our reply 2.2, where we explain in detail the sediment provenance and the individual contributors of terrestrial material to the core sites.

Reviewer #3 (Reviewing the revision): After reading through the revised manuscript and the authors' reply to 2.2, the justification of sea level rise versus riverine material is still not clear to me. While I am only one reader, if I suspect if I am having this confusion, the broad audience of this journal will also struggle with this distinction of the sources of material to the sediment. In the revised results, (lines 103-106) it is stated that the highest rates of sediment accumulation coincide with melt water pulses suggesting that coastal erosion was the main cause of the carbon mobilization. The next paragraph talks about how riverine contributions play an important role. But the rest of the manuscript focuses solely on the coast erosion.

Below are a few suggestions on how you may be able to persuade reader of this assertion.

1) Consider mentioning the magnitude of sea level rise during this time period here and not waiting until line 218 to share this information will help the readers. Not everyone reading this paper will be familiar with that information.

Reply: We changed the lines 104-108 of the revised manuscript (R2) to: "These records of terrigenous material accumulation reach their highest values during times of rapid sea-level rise (Fig. 2b)¹⁴, connected with the global melt water pulses (MWP) centered around 11 and 14 kyrs BP raising sea-level by a total of approximately 80 m. We interpret his temporal coincidence as indication that coastal erosion was the main cause for this permafrost carbon mobilization."

2) I suggest you make the timing difference between the discharge and the accumulation rate clearer. Currently its written that there is a difference in timing (line 127). But I think it could be clearer for your readers if you say that the discharge happened after the peak in accumulation.

Reply: The timing of maximum Amur river discharge (10 kyrs BP) is given in the sentence immediately prior to the one in line 127 of R1 (line 130 in R2). Moreover, in both sentences, reference is made to Figure 2, in which the maxima are clearly evident. Nonetheless, we changed sentence starting in line 130 of R2 to: "However, because of the different timing of the Amur discharge maximum and the peaks in biomarker accumulation rates centered at approximately 14 and 11 kyrs BP, we conclude that river-derived organic matter has only partially contributed to these peaks."

3) A back of the envelope calculation of how much carbon could be mobilized by 80 meters of sea level rise may be useful for readers. For example, if sea level rise rose by 80 meters, it flooded X amount of land and therefore potentially mobilized Z amount of C. And then if this amount of C was compared to contemporary riverine export would be helpful, as it would put the scale of the mobilized C into perspective.

Reply: As the reviewer rightly points out, our model described in lines 224 and following of the revised manuscript (R2) relies on estimating the amount of carbon that could potentially be mobilized by 80 m of sea-level rise. We estimate this to be 129.5 PgC (see line 229 of R2)). In response to reviewers' comments on the original submissions, we refined our estimate of carbon remineralization to 66% of this mobilized carbon (see lines 231-234 of R2). These estimates were clearly described in our R1.

However, it is even more difficult to say how much carbon might have been discharged by the Amur river over the same time period. Based on modern yields of total suspended sediment ($52 * 10^9 \text{ g yr}^{-1}$; Milliman and Farnsworth, 2011) and empirical relations between total sediment yield and carbon yield (Galy et al., 2005), annual carbon yield of the Amur today is estimated to be approximately $5.5 * 10^8 \text{ gCyr}^{-1}$ (compared to an annual yield of the major Arctic rivers of $4\text{-}6 * 10^{12} \text{ gCyr}^{-1}$ (Dittmar and Kattner, 2003)). Our estimate of $1.3 * 10^{17} \text{ gC}$ released by coastal erosion across the deglaciation translates to annual C yields by coastal erosion of $1.9 * 10^{13} \text{ gC yr}^{-1}$. Thus, riverine carbon yield would be 5 (Amur only) to 1 (Arctic rivers) orders of magnitude smaller than that of coastal erosion. However, we consider these numbers highly speculative, as even today's carbon yield is an estimate based on empirical relationships. Past changes in runoff and vegetation (as shown in our paper) likely had strong impacts on the carbon yield of the Amur river. We do not think that such rough estimates should be presented in a scientific paper, and thus refrain from providing the back-of-the envelope calculations as requested by the reviewer.

Otherwise, we are happy to describe in more detail what was done to estimate the flooded shelf area: We tested 2 different bathymetric maps (1. Version 12.1 of a global bathymetry (Smith and Sandwell, 1997) = similar as in Köhler et al., 2014, our ref 3, and 2. ETOPO1, the approach used in Brosius et al (2012), our ref 30, both with 1 min spatial resolution), which slightly differed in the area of flooded Arctic shelf during sea level rise from -130- to -50 m. However, both approaches consistently showed that ~50% of the Arctic shelf area has been flooded during sea-level rise from -130 to -50 m. The total amount of carbon lost from Yedoma during the transition from the glacial to the Holocene amounts to 259 PgC (ref 23), most of which must have been located on the flooded shelf areas. If all carbon on the flooded shelf area (assuming spatially homogenous distribution of carbon containing deposits on the shelf) was mobilized, we estimate a total of ~130 PgC to be mobilized.

To clarify our approach to estimate coastal erosion, we edited lines 225-228 of R2 to read "Considering the regional bathymetry we calculated consistently from two different approaches (similarly as in either ref.3 or ref. 30), that about 50% of the Arctic shelf area was flooded during this time period by this 80 m rise in global sea level."

In light of the substantial uncertainty inherent in our estimate of the flooded shelf area deriving from the differences between the two data sets used, we refrain from providing an estimate of the actual area of the flooded shelf but instead prefer to report fractions of the total area.

Reviewer #3 (original review): Figure 1: I found these maps confusing. The current shoreline was not evident and due to the greyscale contrast of the seafloor, the nearby subduction was a focal point. Please edit these maps to clearly highlight the current sea shore and the study area.

Authors' response: In both panels, the current shoreline is marked (solid line in panel a illustrating the present situation, dashed line in panel b showing the glacial extent of permafrost). Perhaps the resolution of the version the reviewer examined was insufficient. We changed the figure caption accordingly.

Reviewer #3 (Reviewing the revision): When I printed the revised manuscript in black and white, I still found the maps hard to read. It is likely that other readers of this paper will also find it difficult to read the maps. Please try looking at them again in B&W and consider the broad audience who may not be used to looking at maps in this region.

We edited the figure by increasing the thickness of the lines indicating the modern coast line in both panels and think that this will help. We do expect, however, that most readers will inspect the figures in colour and on a computer screen, and we do not anticipate any problems with identifying the coast line in both panels.